# Diffusion Models for Imperceptible and Transferable Adversarial Attack

## Abstract

Many existing adversarial attacks generate $L_p$-norm perturbations on image RGB space. Despite some achievements in transferability and attack success rate, the crafted adversarial examples are easily perceived by human eyes. Towards visual imperceptibility, some recent works explore unrestricted attacks without $L_p$-norm constraints, yet lacking transferability of attacking black-box models. In this work, we propose a novel imperceptible and transferable attack by leveraging both the generative and discriminative power of diffusion models. Specifically, instead of direct manipulation in pixel space, we craft perturbations in the latent space of diffusion models. Combined with well-designed content-preserving structures, we can generate human-insensitive perturbations embedded with semantic clues. For better transferability, we further "deceive" the diffusion model which can be viewed as an implicit recognition surrogate, by distracting its attention away from the target regions. Extensive experiments on various model structures, datasets, and defense methods have demonstrated the superiority of our attack.

## 1 Introduction

Recent years have witnessed remarkable performance exhibited by deep neural networks (DNNs) across a range of domains, including autonomous driving (Feng et al., 2023; Zou et al., 2022), medical image analysis (Zhang et al., 2023; 2022b), remote sensing (Chen et al., 2022a;b), *etc*. Notwithstanding the indisputable advances, early investigations (Szegedy et al., 2013) have elucidated the susceptibility of DNNs to meticulously engineered subversions (hereafter referred to as "adversarial examples"), which may induce grievous mistakes in real-world applications. Moreover, the transferability of these adversarial examples across distinct model architectures (Papernot et al., 2016) poses an even greater hazard to practical implementations. Therefore, it is of the utmost necessity to uncover as many lacunae in machine perception – what may be termed "blind spots" – as can feasibly be achieved, so as to bolster the DNNs' resilience when faced with adversarial challenges.

Compared to white-box attacks (Madry et al., 2018; Goodfellow et al., 2014) that the attacker can access the architecture and parameters of the target model, black-box attacks (Brendel et al., 2018; Narodytska & Kasiviswanathan, 2016; Papernot et al., 2016) can not obtain the target's information and thus are much closer to real-world scenarios. Among black-box directions, we here focus on the transfer-based attacks (Papernot et al., 2016) that directly apply the adversarial examples constructed on a surrogate model to fool the target model. By adopting different optimization strategies (Lin et al., 2020; Dong et al., 2018), designing various loss functions (Lu et al., 2020; Inkawhich et al., 2019), leveraging multiple data augmentations (Long et al., 2022; Xie et al., 2019; Dong et al., 2019), *etc.*, existing approaches have achieved much success and improved the attack's transferability.

$L_p$**-norm based methods**. Most of the above methods adopt $L_p$-norm in RGB color space as an indicator of human perception and constrain the amplitude of the adversarial perturbations under a specific value. Despite the efforts paid, these pixel-based attacks are still easy to be perceived by human eyes, and $L_p$-norm was recently found unsuitable to measure the perceptual distance between two images (Zhao et al., 2020; Johnson et al., 2016). From the examples displayed in Figure 1, the perturbations optimized by $L_p$ loss are noticeable and appear similar to high-frequency noise (indicate overfits on the surrogate model) despite low $L_\infty$ values, which can hinder the transferability to other black-box models (Jia et al., 2022; Sharma et al., 2019) and is easy to be defended against by purification defenses (Nie et al., 2022; Naseer et al., 2020).

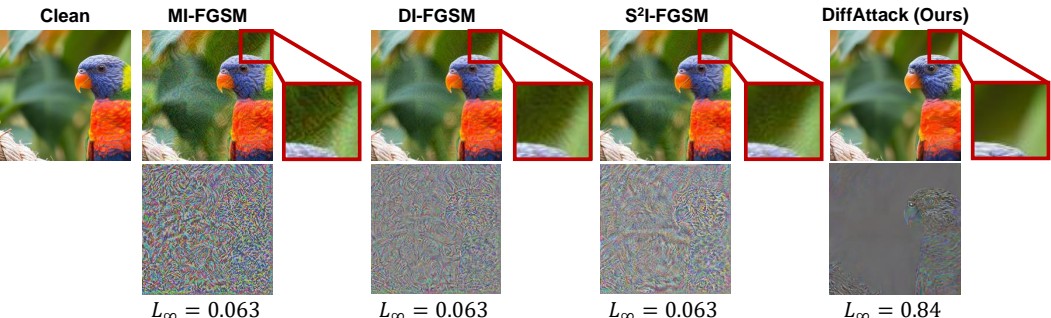

| Clean | MI-FGSM | DI-FGSM | S²I-FGSM | DiffAttack (Ours) |

$L_\infty = 0.063$   $L_\infty = 0.063$   $L_\infty = 0.063$   $L_\infty = 0.84$

Figure 1: **Adversarial perturbations crafted by some attacks.** The second row denotes the difference between the clean image and the adversarial example. Please zoom in for a better view.

**Towards imperceptible attacks**. Recent works (Jia et al., 2022; Yuan et al., 2022; Zhao et al., 2020) explored new ways to deceive human perception without using the $L_p$-norm constraint (*a.k.a.* unrestricted attacks). By applying perturbations on spaces such as object attribute (Jia et al., 2022), color mapping matrix (Yuan et al., 2022), *etc.*, the adversarial examples are well imperceptible despite large $L_p$-norm values in RGB space. Furthermore, these works (Jia et al., 2022; Yuan et al., 2022) revealed that the perturbation generated by unrestricted attacks more focuses on relatively large-scale patterns with high-level semantics, instead of manipulating pixel-level intensity, thus benefiting the attack's transferability to other black-box models and even the defended ones. However, these methods' transferability still lags behind the pixel-based ones.

In this work, we propose a novel unrestricted attack based on diffusion models (Rombach et al., 2022). Instead of manipulating pixels directly, we optimize the latent of an off-the-shelf pretrained diffusion model (Rombach et al., 2022). Besides the basic transferability advantages of high-level perturbations mentioned above, our motivation for introducing the diffusion model into the adversarial attack domain stems primarily from its two beneficial properties. 1) *Good imperceptibility*. Diffusion models, originally designed for image synthesis, tend to generate natural-looking images in line with human perception. This inherent quality aligns well with the imperceptibility requirement of adversarial attacks. Moreover, the iterative denoising process within diffusion models aids in reducing perceptible high-frequency noise. 2) *Approximation of an implicit surrogate*. Despite being initially designed for image synthesis, diffusion models trained on large-scale datasets exhibit a notable discriminative capability (Xu et al., 2023; Clark & Jaini, 2023). This feature enables us to approximate them as implicit surrogate models for transfer-based attacks. Leveraging this "implicit surrogate", we can potentially enhance transferability across different models and defenses. Furthermore, the denoising process of diffusion models, akin to a robust purification defense (Nie et al., 2022), can further bolster the effectiveness of our attack against defensive mechanisms.

To harness the favorable attributes of diffusion models, our work encompasses three key aspects. *Firstly*, we establish a foundational attack framework that initially converts clean images into noise and subsequently introduces modifications in the latent space. This differs from existing image editing techniques (Hertz et al., 2022; Parmar et al., 2023) that manipulate guided text to achieve content editing. Instead, we directly operate on the latents of diffusion models which can significantly enhance attack success. *Secondly*, we propose to deviate the cross-attention maps between text and image pixels, in which way we can transform the diffusion model into an implicit surrogate model that can be practically deceived and attacked. *Finally*, to avoid distorting the initial semantics, specific measures, including self-attention constraint and inversion strength, are considered. We term the proposed unrestricted attack as *DiffAttack*, and our contributions can be summarized as follows:

- We reveal that with its remarkable generative and implicit discriminative capabilities, the diffusion model is a promising foundation for creating adversarial examples that exhibit both high imperceptibility and transferability.

- We propose *DiffAttack*, a novel unrestricted attack where the good properties of diffusion models are leveraged by careful designs. By utilizing the cross- and self-attention maps and attacking the latent of the diffusion model, DiffAttack is both imperceptible and transferable.

- Extensive experiments on a variety of model architectures, datasets, and defense methods (some are presented in the Appendix) have demonstrated the superiorities of our work over the existing attack methods.

## 2 RELATED WORKS

**Transferable Attacks.** Transfer-based attacks resort to a surrogate model and rely on the cross-model transferability of adversarial examples for achieving black-box attacks. By crafting perturbations to the surrogate model, they expect these adversarial examples can also have a good effect on the target model. To enhance the generalization of adversarial examples crafted on surrogate models, previous works put a lot of effort into keeping perturbations from getting stuck in a model-specific local optimum that overfits the surrogate model and cannot transfer well to other methods. Xiong et al. (2022); Wang & He (2021) adopted the straightforward strategy of *model ensembles* to attack as many models as possible by finding an optimum updated direction. Long et al. (2022); Xie et al. (2019); Dong et al. (2019) proposed to leverage *data augmentations* to diversify the inputs, which ensures the attack robustness under different scenarios. Lu et al. (2020); Naseer et al. (2019); Inkawhich et al. (2019) applied *loss functions* on the feature space which demonstrated good performance on black-box targets. Lin et al. (2020); Dong et al. (2018) combined momentum into *optimization schedules* to help jump out of local optimum. Despite the much improvement in the transferability, these works mostly conduct attacks with $L_p$-norm constraint on RGB pixel space, resulting in high-frequency noises and patterns (see Figure 1) which, though hold a relatively low value on $L_p$-norm, are easy to be perceived by humans. In contrast, our *DiffAttack* perturbs the latent in diffusion models, achieving good imperceptibility together with excellent transferability across various black-box models.

**Unrestricted Attacks.** Since $L_p$-norm in RGB space was found not ideal for measuring the perceptual distance (Jia et al., 2022; Yuan et al., 2022), recent research turns to unconstraint and proposes unrestricted but imperceptible attacks. Zhao *et al.* (Zhao et al., 2020) adopted CIEDE2000 which can better indicate the perceptual color loss. Qiu *et al.* (Qiu et al., 2020) and Jia *et al.* (Jia et al., 2022) achieve imperceptibility by modifying the attributes of the images, especially human faces. Yuan *et al.* (Yuan et al., 2022) constructed a color distribution library, which is used to find a successful distribution for adversarial attacks. However, despite their good imperceptibility, these methods generally cannot compete with the aforementioned pixel-based methods in terms of transferability. Our work also falls in this direction but achieves better transferability and imperceptibility, and is the first to explore the strength of diffusion models in crafting unrestricted attacks.

**Diffusion Models.** Recently, diffusion models (Sohl-Dickstein et al., 2015; Ho et al., 2020) have attracted extensive attention and shown their fabulous power. Images are first converted into purely Gaussian noise in the *forward process* and then a U-Net structure is trained to predict the added noise in each timestep of the *reversed process*. Being trained on large numbers of data, the diffusion models (Saharia et al., 2022a; Ramesh et al., 2022; Rombach et al., 2022) can either generate high-quality images from randomly sampled noise, or more specific ones that follow the guidance of text prompt. Due to its significant performance, the diffusion model has also diffused to other areas, such as image inpainting (Li et al., 2022; Xie et al., 2022), image super-resolution (Saharia et al., 2022b), real image editing (Parmar et al., 2023; Mokady et al., 2022), *etc.* Recent work further showed that the pretrained diffusion models can be taken as good recognition models (Xu et al., 2023; Clark & Jaini, 2023) and denoisers (Nie et al., 2022). Despite the many applications mentioned above, the potential of diffusion models in the adversarial attack field remains underexplored.

## 3 METHOD

### 3.1 PROBLEM FORMULATION

Given a clean image $x$ and its corresponding label $y$, attackers aim to craft perturbations that can deviate the decision of a classifier $F_\theta$ ($\theta$ denotes the model's parameters) from correct to wrong:

$$F_\theta(\text{Attack}(\text{x}; \text{G}_\phi)) = F_\theta(x') \neq y \tag{1}$$

where $\text{Attack}(\cdot)$ is the attack approach and $x'$ is the crafted adversarial example. Since $F_\theta$ is inaccessible in black-box scenarios, the adversarial examples are crafted on a surrogate model $G_\phi$.

Different from previous pixel-based attacks (Dong et al., 2018; Long et al., 2022) that apply $L_p$-norm constraints on pixel values ($\|\epsilon\|_p < c$, where $\epsilon$ is the perturbation and $c$ is a hyperparameter), we impose perturbations in the latent space of the diffusion model and rely on properties of the diffusion model to achieve visually natural and successful attacks. We describe our design in detail below.

### 3.2 BASIC FRAMEWORK

We display in Figure 2 the whole framework of *DiffAttack*, where we adopt the open-source Stable Diffusion (Rombach et al., 2022) that pretrained on extremely massive text-image pairs. Since adver-

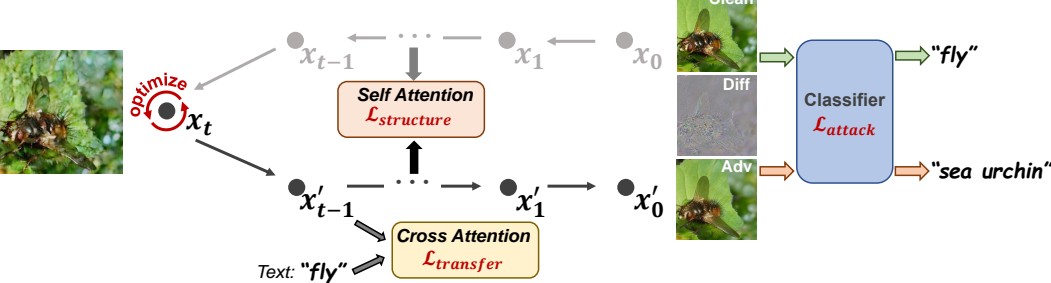

Figure 2: **Framework of *DiffAttack*.** We adopt Stable Diffusion (Rombach et al., 2022) and leverage DDIM Inversion (Song et al., 2021) to convert the clean image into the latent space. The latent is optimized to deceive the classifier. The cross-attention maps are leveraged to "deceive" the diffusion model, and we use self-attention maps to preserve the structure. For simplicity, we here do not display the unconditional optimization, whose details can be referred to Section 3.4.

sarial attacks aim to fool the target model by perturbing the initial image, they can be approximated as a special kind of real image editing. Inspired by recent diffusion editing approaches (Couairon et al., 2022; Mokady et al., 2022; Parmar et al., 2023), our framework leverages the DDIM Inversion technology (Song et al., 2021), where the clean image is mapped back into the diffusion latent space by reversing the deterministic sampling process:

$$x_t = \text{Inverse}(x_{t-1}) = \underbrace{\text{Inverse} \circ \cdots \circ \text{Inverse}}_{t}(x_0) \tag{2}$$

where $\text{Inverse}(\cdot)$ denotes the DDIM Inversion operation (Please see Appendix B for details. In Eq. 2, we ignore the autoencoder stage of the Stable Diffusion (Rombach et al., 2022) for simplicity). We apply the inversion for several timesteps from $x_0$ (the initial image) to $x_t$. A high-quality reconstruction of $x_0$ can then be expected if we conduct the deterministic denoising process from $x_t$ (Dhariwal & Nichol, 2021; Song et al., 2021).

Many of the existing image editing approaches (Couairon et al., 2022; Mokady et al., 2022) proposed to modify text embeddings for image editing, through which way, the image latent $x_t$ can gradually shift to the target semantic space during the iterative denoising process with the text guidance. However, in our explorations (see Appendix C), we found that the perturbations on the guided text embeddings would be hard to work on the other black-box models, leading to weak transferability. Therefore, different from the editing approaches, we here propose to directly perturb the latent $x_t$:

$$\arg\min_{x_t} \mathcal{L}_{attack} = -J(x', y; G_\phi), \quad \text{where } x' = x'_0 = \underbrace{\text{Denoise} \circ \cdots \circ \text{Denoise}}_{t}(x_t) \tag{3}$$

where $J(\cdot)$ is the cross-entropy loss and $\text{Denoise}(\cdot)$ denotes the diffusion denoising process. An initial concern might arise regarding the potential generation of unnatural results using this straightforward method. However, we can observe in Figure 1 that the difference is almost indistinguishable between the image reconstructed from the perturbed latent and the initial clean one. Furthermore, we can notice that the difference image encapsulates numerous high-level semantic cues, as opposed to the high-frequency noise typically associated with pixel-based attacks. We attribute this phenomenon to the denoising process of the diffusion model, which effectively reduces perceptible high-frequency noise. These semantically rich perturbations can not only enhance the imperceptibility but also benefit the attack's transferability (Jia et al., 2022).

### 3.3 "DECEIVE" DIFFUSION MODEL

According to the research by Nie et al. (2022), the reversed process of the diffusion model is a strong adversarial purification defense. Thus, our perturbed latent will experience purification before being decoded to the final image, which then ensures the naturalness of crafted adversarial examples and also the robustness towards other purification denoises (see Section 4.2.2).

In addition to leveraging the denoising component, we here go a further step to enhance our attack's transferability. Given an image and its corresponding caption, we can see from Figure 3 that in the reconstruction process of the inversed latent, the cross-attention maps display a strong relationship between the guided text and the image pixels, which demonstrates the potential recognition capability of pretrained diffusion models. Such a relationship is also verified by Hertz *et al.* (Hertz et al., 2022) and its recognition power recently has been leveraged on the downstream tasks (Xu et al., 2023; Clark

& Jaini, 2023). Thus, the diffusion model that is trained on massive data can be approximated as an implicit recognition model, and our motivation is that, if our crafted attacks can "deceive" this model, we may expect an improvement of the transferability to other black-box models.

Denote $C$ as the caption of the clean image, which we set to the groundtruth category's name (we can also simply use the predicted category of $G_\phi$, and thus not rely on true labels). We accumulate the cross-attention maps between image pixels and $C$ in all the denoising steps and get the average. To "deceive" the pretrained diffusion model, we propose to minimize the following formula:

$$\arg\min_{x_t} \mathcal{L}_{transfer} = \text{Var}(\text{Average}(\text{Cross}(x_t, t, C; \text{SDM}))) \tag{4}$$

where $\text{Var}(\cdot)$ calculates the input's variance, $\text{Cross}(\cdot)$ denotes the accumulation of all the cross-attention maps in the denoising process, and SDM is the Stable Diffusion. The insight is to distract the diffusion model's attention from the labeled objects. By evenly distributing attention to each pixel, we can disrupt the original semantic relationship, ensuring that our crafted adversarial examples well "deceive" the diffusion model. With such a design, DiffAttack exhibits an *implicit ensemble characteristic*. Note that it differs significantly from typical explicit ensemble attacks (Tramèr et al., 2018), about which we give a detailed analysis in Appendix K.

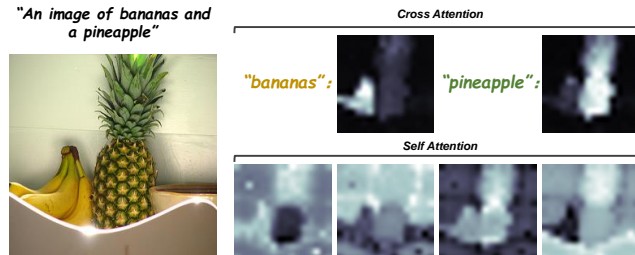

Figure 3: **Visualization of cross- and self- attention maps.** There is a strong relationship between text and pixels in cross-attention, while self-attention can well reveal structure.

### 3.4 PRESERVE CONTENT STRUCTURE

As mentioned in Section 3.2, our unrestricted attack can be approximated as an image editing approach, thus the change of the content structure is unavoidable. If the degree of the changes is not under control, the resulting adversarial examples may lose most semantics of the initial clean image (see Figure 5), which loses the significance of the adversarial attacks and is not what we want. Therefore, we here preserve the content structure mainly from two perspectives.

**Self-Attention Control.** Researches by Tumanyan et al. (2022); Shechtman & Irani (2007) have discovered that the self-similarly-based descriptors can capture structural information while ignoring image appearance. Along with this idea, we can observe from Figure 3 that the self-attention in the diffusion models also has that property embedded in it, which is in contrast to cross attention that mainly focuses on high-level semantics. Therefore, we propose to leverage the self-attention maps for structure retention. We set a copy $x_{t(fix)}$ of the inversed latent which is fixed without perturbations. By respectively calculating the self-attention maps (denoted as $S_{t(fix)}$ and $S_t$) of $x_{t(fix)}$ and $x_t$, we force $S_t$ to get close to $S_{t(fix)}$ as follows:

$$\arg\min_{x_t} \mathcal{L}_{structure} = \|S_t - S_{t(fix)}\|_2^2 \tag{5}$$

Similar to Eq. 4, we here apply the self-attention constraint to all the denoising steps. Since $x_{t(fix)}$ reconstructs the initial clean image well (Song et al., 2021), we can in this way preserve the structure.

**Inversion Strength Trade-off.** With DDIM Inversion strength increased, the latent $x_t$ will get closer to pure Gaussian distribution and the perturbations on it may cause serious distortion due to influence on more denoising steps (see Figure 5). Whereas, a limited inversion cannot provide enough space for attacking, since the latent image prior is too strong. The inversion strength is a trade-off between imperceptibility and the attack success. Recent work (Meng et al., 2021) has found that the diffusion models tend to add coarse semantic information (*e.g.*, layout) in the early denoising steps while more fine details in the later steps. Thus, we control the inversion at the back of the denoising process for retention of high-level semantics, and reduce the total DDIM sample steps for more editing space.

Besides the above operations, we also adopt the approach of Mokady et al. (2022) to get a good initial reconstruction by optimizing unconditional embeddings. Details can be found in their source paper.

In general, the final objective function of *DiffAttack* is as follows, where $\alpha$, $\beta$, and $\gamma$ represent the weight factors of each loss:

$$\arg\min_{x_t} \mathcal{L} = \alpha\mathcal{L}_{attack} + \beta\mathcal{L}_{transfer} + \gamma\mathcal{L}_{structure} \tag{6}$$

## 4 EXPERIMENTS

### 4.1 EXPERIMENTAL SETUP

**Datasets.** Following the previous methods (Long et al., 2022; Yuan et al., 2022; Zhao et al., 2020), we evaluate the performance of our attack on the development set of ImageNet-Compatible Dataset[1], which consists of 1,000 images with size 299×299×3. Considering that the Stable Diffusion cannot handle the original input size of the ImageNet-Compatible Dataset, we focused on a resized version of 224×224×3 in all the experiments. DiffAttack also generalizes well to other datasets. Please refer to Appendix I where we conduct further experiments on CUB-200-2011 (Wah et al., 2011) and Stanford Cars (Krause et al., 2013).

**Models.** We evaluate the transferability of the attacks across a variety of network structures, including CNNs, Transformers, and MLPs. For CNNs, we adopt normally trained models including ConvNeXt (Liu et al., 2022), ResNet-50 (Res-50) (He et al., 2016), VGG-19 (Simonyan & Zisserman, 2014), Inception-v3 (Inc-v3) (Szegedy et al., 2016), and MobileNet-v2 (Mob-v2) (Sandler et al., 2018). For Transformers, we consider normally trained ViT-B/16 (ViT-B) (Dosovitskiy et al., 2021), Swin-B (Liu et al., 2021), DeiT-B and DeiT-S (Touvron et al., 2021). For MLPs, we adopt normally trained Mixer-B/16 (Mix-B) and Mixer-L/16 (Mix-L) (Tolstikhin et al., 2021). Furthermore, we also consider various defense methods, including DiffPure (Nie et al., 2022), RS (Jia et al., 2020), R&P (Xie et al., 2018), HGD (Liao et al., 2018), NIPS-r3 (Thomas & Elibol, 2017), NRP (Naseer et al., 2020), and adversarially trained models (Adv-Inc-v3 (Kurakin et al., 2018), Inc-v3$_{ens3}$, Inc-v3$_{ens4}$, and IncRes-v2$_{ens}$ (Tramèr et al., 2018)).

**Implementation Details.** We leverage DDIM (Song et al., 2021) as the sampler of the Stable Diffusion (Rombach et al., 2022). The number of steps is set to 20 and we apply 5 DDIM Inversion steps of the initial clean image. In the inversion process, the guidance scale is set to 0, while in the denoising process, we set it to 2.5. For optimizing the latent $x_t$, we adopt AdamW (Loshchilov & Hutter, 2019) with the learning rate set to $1e^{-2}$ and the iterations set to 30. The weight factors $\alpha$, $\beta$, $\gamma$ in Eq. 6 are set to 10, 10000, 100 respectively. All experiments are run on a single RTX 3090 GPU.

**Evaluation Metrics.** We adopt top-1 accuracy to evaluate the performance of the attack methods and leverage Frechet Inception Distance (FID) (Heusel et al., 2017) as the indicator of the human imperceptibility of the crafted adversarial examples. A full-referenced metric, LPIPS (Zhang et al., 2018), is also used to assess the perceptual differences which can be found in Appendix H, I, J.

### 4.2 COMPARISONS

#### 4.2.1 RESULTS ON NORMALLY TRAINED MODELS

Here, we compared the performance of *DiffAttack* on normally trained models with other transfer-based black-box attacks. We select five pixel-based attacks (MI-FGSM (Dong et al., 2018), DI-FGSM (Xie et al., 2019), TI-FGSM (Dong et al., 2019), PI-FGSM (Gao et al., 2020), S$^2$I-FGSM (Long et al., 2022)) and two unrestricted attacks (PerC-AL (Zhao et al., 2020), NCF (Yuan et al., 2022)). Except that the resolution is changed to 224×224×3, the implementations of these methods follow their original optimal settings (See Appendix F for details). We craft the adversarial examples via Res-50, VGG-19, Mov-v2, Inc-v3, ConvNeXt, and Swin-B (Performance on more surrogate models can be found in Appendix H). The transferability of different attack methods is displayed in Table 1.

From the results, we can observe that *DiffAttack* can achieve the best transferability across a variety of model structures, while other unrestricted attacks (PerC-AL and NCF) usually fail to compete with pixel-based attacks. In some architectures such as VGG-19 and Mob-v2, our method can even outperform the second-best method by nearly 10 points (38.2% vs. 49.0%, 40.5% vs. 49.9%). It may be noticed that our method fails to compete with MI-FGSM and PI-FGSM under Inc-v3 structure, however, from the FID results, we have a large advantage over them (about 20 or more points lower).

For the imperceptibility of the crafted adversarial examples, we can observe that PerC-AL has always achieved the best FID result. However, it can hardly deceive other black-box models and achieves the worst transferability where the accuracy value of AVG(w/o self) is very close to the clean image. Therefore, we choose to ignore the PerC-AL results here, and our method achieves the best performance. We turn the color of the best performance in Table 1 to red for a better view.

---

[1] https://github.com/cleverhans-lab/cleverhans/tree/master/cleverhans_v3.1.0/examples/nips17_adversarial_competition/dataset.

Table 1: **Transferability and imperceptibility comparisons on normally trained models.** We report top-1 accuracy(%) of each method. "S." denotes surrogate models while "T." denotes target models. For white-box attacks (surrogate model same as target), we set the background to gray. "AVG(w/o self)" denotes the average accuracy on all the models except the one that same as the surrogate. "FID" is calculated between the 1,000 images of the ImageNet-Compatible dataset with the ImageNet validation set. The best result is bolded, and the second-best result is underlined.

| T. S. / Attacks | CNNs | | | | | Transformers | | | | MLPs | | AVG(w/o self) | FID |
|---|---|---|---|---|---|---|---|---|---|---|---|---|---|
| | Res-50 | VGG-19 | Mob-v2 | Inc-v3 | ConvNeXt | ViT-B | Swin-B | DeiT-B | DeiT-S | Mix-B | Mix-L | | |
| Clean | 92.7 | 88.7 | 86.9 | 80.5 | 97.0 | 93.7 | 95.9 | 94.5 | 94.0 | 82.5 | 76.5 | 89.4 | 57.8 |
| **Res-50** | | | | | | | | | | | | | |
| MI-FGSM | 0 | 19.9 | 20.2 | 28.9 | 57.8 | 67.3 | 67.0 | 72.4 | 67.0 | 52.2 | 45.4 | 49.8 | 81.2 |
| DI-FGSM | 0 | 21.2 | 20.5 | 34.5 | 71.6 | 82.0 | 75.3 | 80.5 | 76.0 | 61.3 | 56.8 | 58.0 | 85.3 |
| TI-FGSM | 0 | 42.4 | 37.1 | 46.0 | 83.6 | 81.6 | 83.7 | 84.5 | 79.0 | 66.0 | 61.7 | 66.6 | 66.0 |
| PI-FGSM | 0 | 14.1 | 15.0 | 24.0 | 72.5 | 65.3 | 77.5 | 76.7 | 65.0 | 50.5 | 43.8 | 50.5 | 97.9 |
| $S^2$I-FGSM | 0 | 9.2 | 6.6 | 18.6 | 44.1 | 63.9 | 52.0 | 65.9 | 59.0 | 45.6 | 44.3 | 40.9 | 79.8 |
| PerC-AL | 6.5 | 83.1 | 80.2 | 76.4 | 96.0 | 93.9 | 94.8 | 94.4 | 93.0 | 81.6 | 75.1 | 86.8 | 58.2 |
| NCF | 11.3 | 30.5 | 30.3 | 52.6 | 78.3 | 65.7 | 76.8 | 75.1 | 67.0 | 53.7 | 47.6 | 57.8 | 70.9 |
| DiffAttack(Ours) | 3.7 | 24.4 | 22.9 | 31.0 | 41.0 | 48.8 | 43.8 | 49.5 | 45.0 | 42.9 | 42.2 | **39.2** | 62.6 |
| **VGG-19** | | | | | | | | | | | | | |
| MI-FGSM | 22.7 | 0 | 15.4 | 33.5 | 53.2 | 73.2 | 63.3 | 74.7 | 68.0 | 54.3 | 48.6 | 50.6 | 82.4 |
| DI-FGSM | 32.2 | 0 | 23.9 | 46.5 | 67.2 | 84.7 | 71.9 | 84.8 | 80.0 | 65.7 | 60.9 | 61.8 | 70.9 |
| TI-FGSM | 44.5 | 0 | 32.8 | 47.4 | 77.8 | 81.4 | 79.3 | 83.6 | 79.0 | 64.9 | 60.3 | 65.1 | 66.6 |
| PI-FGSM | 22.7 | 0 | 16.4 | 29.8 | 68.3 | 68.0 | 75.7 | 79.5 | 68.0 | 50.9 | 41.8 | 52.1 | 96.4 |
| $S^2$I-FGSM | 17.9 | 0 | 11.3 | 31.8 | 49.5 | 74.1 | 57.9 | 76.0 | 68.0 | 52.6 | 50.8 | 49.0 | 82.9 |
| PerC-AL | 87.5 | 4.6 | 79.0 | 76.1 | 95.1 | 94.2 | 94.0 | 94.3 | 93.0 | 81.3 | 75.1 | 87.0 | 57.9 |
| NCF | 38.3 | 6.8 | 31.5 | 52.4 | 80.5 | 67.5 | 77.6 | 77.4 | 71.0 | 53.5 | 47.2 | 59.7 | 70.4 |
| DiffAttack(Ours) | 21.1 | 2.7 | 19.4 | 29.7 | 43.1 | 52.9 | 41.6 | 51.3 | 45.0 | 39.6 | 38.5 | **38.2** | 63.9 |
| **Mob-v2** | | | | | | | | | | | | | |
| MI-FGSM | 26.4 | 18.7 | 0 | 31.0 | 62.0 | 69.5 | 65.2 | 71.6 | 63.0 | 46.9 | 44.4 | 49.9 | 76.4 |
| DI-FGSM | 28.7 | 18.9 | 0 | 33.9 | 73.4 | 79.9 | 71.4 | 79.6 | 75.0 | 57.7 | 57.1 | 57.6 | 78.6 |
| TI-FGSM | 47.2 | 37.9 | 0 | 45.2 | 83.0 | 79.9 | 80.9 | 81.8 | 76.0 | 61.7 | 58.3 | 65.1 | 65.6 |
| PI-FGSM | 21.1 | 13.3 | 0 | 27.6 | 74.4 | 65.3 | 77.0 | 77.4 | 66.0 | 49.7 | 41.5 | 51.4 | 98.7 |
| $S^2$I-FGSM | 21.0 | 13.4 | 0 | 27.2 | 64.3 | 74.1 | 62.6 | 75.2 | 68.0 | 51.4 | 48.3 | 50.5 | 79.4 |
| PerC-AL | 88.2 | 84.2 | 5.9 | 76.8 | 96.2 | 93.9 | 94.2 | 94.3 | 94.0 | 81.2 | 74.3 | 87.7 | 58.1 |
| NCF | 36.0 | 29.4 | 7.4 | 51.9 | 77.4 | 67.2 | 76.1 | 76.1 | 68.0 | 54.9 | 48.3 | 58.6 | 69.7 |
| DiffAttack(Ours) | 23.6 | 23.4 | 1.8 | 31.6 | 50.3 | 51.4 | 45.8 | 53.4 | 46.0 | 38.5 | 40.8 | **40.5** | 62.9 |
| **Inc-v3** | | | | | | | | | | | | | |
| MI-FGSM | 42.8 | 36.6 | 34.4 | 0 | 79.8 | 75.3 | 79.4 | 78.6 | 73.0 | 56.5 | 48.9 | 60.6 | 80.5 |
| DI-FGSM | 61.7 | 57.4 | 51.9 | 0.2 | 89.9 | 84.6 | 86.8 | 86.7 | 82.0 | 68.4 | 62.3 | 73.2 | 67.1 |
| TI-FGSM | 76.0 | 70.1 | 66.7 | 0.1 | 93.8 | 88.7 | 91.2 | 89.7 | 88.0 | 73.8 | 66.8 | 80.5 | 62.8 |
| PI-FGSM | 37.9 | 22.4 | 28.4 | 0 | 81.0 | 74.3 | 83.0 | 81.9 | 72.0 | 57.1 | 45.8 | 58.4 | 92.5 |
| $S^2$I-FGSM | 52.3 | 47.8 | 43.3 | 0 | 86.3 | 80.8 | 84.1 | 83.8 | 78.0 | 63.5 | 57.3 | 67.8 | 72.5 |
| PerC-AL | 90.8 | 87.0 | 85.8 | 7.7 | 97.5 | 93.6 | 95.1 | 94.2 | 94.0 | 81.5 | 75.3 | 89.4 | 58.4 |
| NCF | 52.6 | 45.8 | 46.2 | 17.4 | 85.7 | 75.9 | 83.4 | 82.7 | 76.0 | 61.1 | 52.9 | 66.2 | 66.7 |
| DiffAttack(Ours) | 59.5 | 55.6 | 55.4 | 13.9 | 76.9 | 75.2 | 72.8 | 74.0 | 71.0 | 58.9 | 54.7 | 65.4 | 62.3 |
| **ConvNeXt** | | | | | | | | | | | | | |
| MI-FGSM | 34.5 | 22.4 | 26.5 | 41.9 | 0 | 63.4 | 18.0 | 56.5 | 56 | 41.4 | 40.2 | 40.1 | 84.5 |
| DI-FGSM | 33.6 | 24.3 | 29.8 | 46.6 | 0 | 71.0 | 18.8 | 62.2 | 64.0 | 49.6 | 46.7 | 44.6 | 79.6 |
| TI-FGSM | 50.7 | 37.3 | 41.1 | 51.8 | 0 | 70.9 | 38.8 | 68.6 | 69.0 | 52.3 | 47.2 | 52.7 | 73.5 |
| PI-FGSM | 23.6 | 14.2 | 17.1 | 22.4 | 0 | 43.0 | 37.2 | 48.7 | 43.0 | 33.2 | 31.7 | 31.4 | 101.8 |
| $S^2$I-FGSM | 13.6 | 9.6 | 11.9 | 20.2 | 0 | 35.4 | 4.2 | 31.0 | 31.0 | 23.2 | 25.6 | 20.5 | 99.4 |
| PerC-AL | 89.0 | 84.5 | 84.0 | 77.5 | 88.9 | 92.8 | 90.0 | 92.4 | 92.0 | 79.3 | 74.5 | 85.6 | 57.7 |
| NCF | 47.1 | 41.4 | 39.2 | 54.7 | 41.4 | 61.6 | 63.9 | 64.8 | 62.0 | 52.2 | 47.8 | 53.5 | 67.0 |
| DiffAttack(Ours) | 20.9 | 24.8 | 21.8 | 25.8 | 1.9 | 26.7 | 11.4 | 21.6 | 24.0 | 21.7 | 24.0 | 22.2 | 73.3 |
| **Swin-B** | | | | | | | | | | | | | |
| MI-FGSM | 55.7 | 42.3 | 42.7 | 55.2 | 42.5 | 70.6 | 0.9 | 64.2 | 64.0 | 52.4 | 47.9 | 53.7 | 72.8 |
| DI-FGSM | 52.7 | 43.0 | 44.5 | 56.4 | 33.9 | 66.6 | 2.7 | 57.2 | 58.0 | 52.4 | 50.8 | 51.5 | 65.7 |
| TI-FGSM | 71.9 | 61.7 | 56.9 | 60.2 | 66.0 | 76.3 | 1.9 | 72.2 | 72.0 | 61.2 | 56.9 | 65.6 | 65.9 |
| PI-FGSM | 38.3 | 21.6 | 25.8 | 35.7 | 54.8 | 48.4 | 0.6 | 52.4 | 47.0 | 43.5 | 38.5 | 40.6 | 89.7 |
| $S^2$I-FGSM | 47.4 | 37.8 | 35.4 | 45.3 | 26.8 | 48.5 | 1.0 | 46.2 | 45.0 | 39.3 | 39.0 | 41.1 | 68.2 |
| PerC-AL | 92.2 | 87.4 | 85.5 | 78.5 | 94.6 | 94.0 | 6.3 | 94.1 | 93.0 | 81.4 | 75.5 | 87.6 | 57.9 |
| NCF | 49.5 | 44.9 | 44.9 | 60.5 | 70.1 | 63.7 | 36.9 | 66.0 | 63.0 | 51.7 | 49.1 | 56.3 | 65.5 |
| DiffAttack(Ours) | 43.5 | 42.1 | 40.7 | 41.4 | 34.0 | 39.0 | 9.9 | 35.0 | 37.0 | 37.7 | 37.4 | **38.8** | 65.5 |

In Figure 4, we visualize the adversarial examples crafted by different attack approaches. We can see that our attack is much more imperceptible compared with MI-FGSM, DI-FGSM, TI-FGSM, PI-FGSM, and $S^2$I-FGSM, where there is high-frequency noise that can be perceived easily. Compared with NCF, *DiffAttack* is more natural in color space. For PerC-AL, although the attack can hard to be perceived, its transferability is the worst as mentioned above. Thus, our method's superiority is well verified. More visualizations can be found in Appendix O.

Besides the above iterative approaches, there exists another category of attacks known as GAN-based attacks (Poursaeed et al., 2018). These attacks do not directly optimize perturbations but instead focus on training a GAN generator. While DiffAttack fundamentally belongs to the iterative approach category, we conduct a comprehensive comparison with these GAN-based attacks in Appendix J, where DiffAttack consistently outperforms them. Additionally, to strengthen the potential of DiffAttack, we compare it with ensemble attacks, which are more powerful, in Appendix G and K.

### 4.2.2 RESULTS ON DEFENSE APPROACHES

To further verify the robustness of each attack method, we evaluate the performance of the crafted adversarial examples on defense approaches. Following Yuan et al. (2022); Long et al. (2022), we consider both input preprocessing defenses (Jia et al., 2020; Xie et al., 2018; Liao et al., 2018; Thomas & Elibol, 2017; Naseer et al., 2020) and adversarially trained models (Kurakin et al., 2018; Tramèr et al., 2018) (see Section 4.1). We further consider the recent DiffPure defense (Nie et al., 2022)

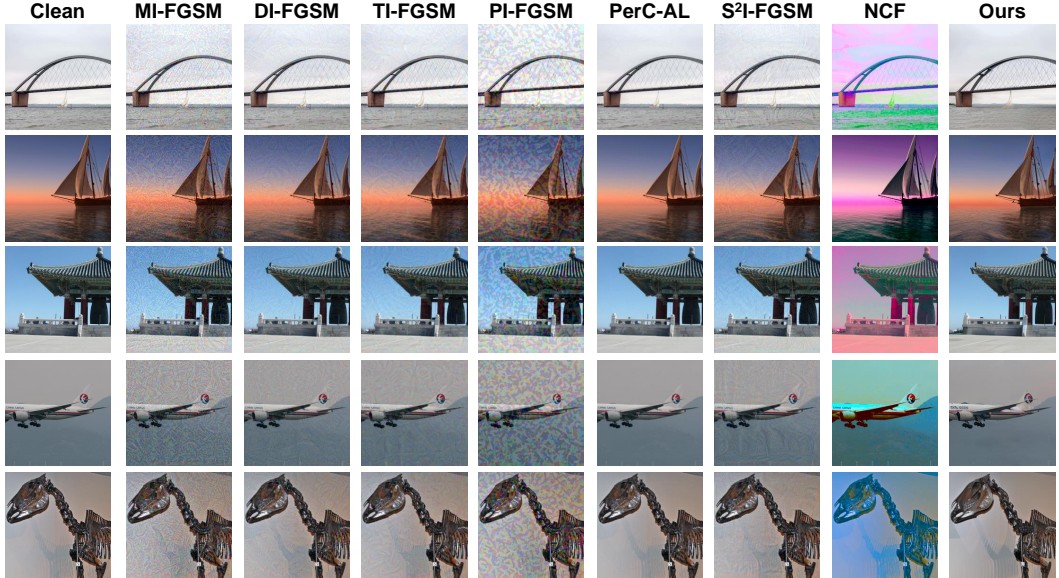

| Clean | MI-FGSM | DI-FGSM | TI-FGSM | PI-FGSM | PerC-AL | S²I-FGSM | NCF | Ours |

Figure 4: **Visual comparisons among different attacks.** Please zoom in for a better view.

Table 2: **Robustness on defense approaches.** We report top-1 accuracy(%) of each method. "A." denotes attack methods while "D." denotes defense approaches. "Inc-v3$_{normal}$" denotes the accuracy on normally trained Inc-v3. For NRP and DiffPure, we display the accuracy differences after purification. The best result is bolded, and the second-best result is underlined.

| D. / A. | HGD | R&P | NIP-r3 | RS | Adv-Inc-v3 | Inc-v3$_{ens3}$ | Inc-v3$_{ens4}$ | IncRes-v2$_{ens}$ | Inc-v3$_{normal}$ | NRP | DiffPure |
|---|---|---|---|---|---|---|---|---|---|---|---|
| MI-FGSM | 77.9 | 76.8 | 65.2 | 62.7 | 51.1 | 49.8 | 53.7 | 70.5 | 0 | +5.9 | +38.4 |
| DI-FGSM | 80.5 | 83.8 | 79.4 | 68.7 | 64.2 | 58.5 | 61.5 | 74.9 | 0.2 | +22.9 | +52.4 |
| TI-FGSM | 84.7 | 86.1 | 87.0 | 69.4 | 66.1 | 62.4 | 64.5 | 76.8 | 0.1 | +25.8 | +55.5 |
| PI-FGSM | 73.4 | 68.6 | **57.2** | 37.1 | **42.3** | 45.0 | 44.6 | 62.0 | 0 | +7.7 | +21.5 |
| S²I-FGSM | 72.5 | 76.5 | 73.3 | 65.0 | 51.8 | 47.0 | 52.2 | 67.7 | 0 | +3.2 | +47.0 |
| PerC-AL | 95.6 | 94.4 | 96.7 | 74.2 | 80.8 | 76.7 | 75.4 | 88.6 | 7.7 | +56.8 | +55.9 |
| NCF | 71.1 | 66.4 | 74.6 | **29.0** | 48.8 | 47.2 | 49.0 | 60.5 | 17.4 | +11.0 | +14.8 |
| DiffAttack(Ours) | **62.0** | **65.5** | 70.0 | 52.8 | 46.0 | **43.8** | **43.1** | **58.3** | 13.9 | **+2.3** | **+13.9** |

to better demonstrate our superiority. We take Inc-v3 as an example surrogate model and all the adversarial examples are crafted from it. For NRP and DiffPure, we set the target model as Inc-v3 itself, thus better revealing the robustness. For other defenses, the target models are the same as the official papers. We display the results in Table 2.

From the results, we can see that our method can achieve good robustness and outperform other methods when some defenses are applied. For the adversarial purification defenses, it can be seen that the attack success of our attack has the least change compared with other ones, which does verify the robustness of *DiffAttack* and the effectiveness of our designs in Section 3.3.

### 4.3 ABLATION STUDIES

In Table 3, we ablate the designs mentioned in Section 3.3. The adversarial examples are crafted on Inc-v3. We can observe that with the loss in Eq. 4 added, the attack success improves, verifying our design's effectiveness. It can also be noted that prompt guidance is important for transferability, which we attribute to the fact that prompts can help guide the attack on the target objects. Moreover, as stated in Section 1, both

Table 3: **Transferability.**

| Prompt Guidance | Diffusion Deception | AVG (w/o self) |
|---|---|---|
| ✗ | ✗ | 70.0 |
| ✓ | ✗ | 66.5 |
| ✓ | ✓ | 65.4 |

Table 4: **Imperceptibility.**

| Steps Inversed | Self-Attn Control | Initial Recon | FID |
|---|---|---|---|
| 10 | ✗ | ✗ | 97.9 |
| 5 | ✗ | ✗ | 66.7 |
| 5 | ✓ | ✗ | 63.5 |
| 5 | ✓ | ✓ | 62.3 |

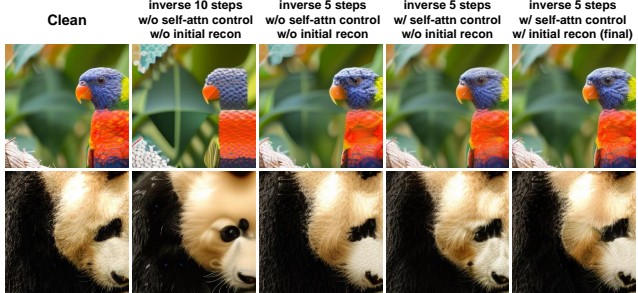

| Clean | inverse 10 steps w/o self-attn control w/o initial recon | inverse 5 steps w/o self-attn control w/o initial recon | inverse 5 steps w/ self-attn control w/o initial recon | inverse 5 steps w/ self-attn control w/ initial recon (final) |

Figure 5: **Visualization of imperceptibility ablations.** Please zoom in for a better view.

our latent perturbation approach and the denoising process in diffusion models also contribute to transferability. A more detailed ablation study on this aspect is provided in Appendix N. Results in Table 4 verify the effectiveness of our designs for structure retention. With the inversion strength and self-attention controlled, the FID result gradually improves. We also visualize the structure ablation in Figure 5, which can display the visual improvement obviously. It can be seen that the control of inversion strength helps a lot preserve the structure, and the usage of self-attention maps can ensure better texture. For more ablation studies on parameter settings, please refer to Appendix O.

## 5 DISCUSSIONS AND OUTLOOKS

Besides the designs outlined in Section 3, we have explored other strategies to enhance imperceptibility and transferability during the exploration of diffusion-based adversarial attacks. While these exploratory endeavors yielded limited success, we deem it valuable to provide an in-depth discussion, as they may contribute to future research. Detailed insights are presented in Appendices D and E.

We are also pleased to observe the rapid growth of concurrent research in diffusion-based attacks, underscoring the potential of this direction. For a comprehensive overview, we briefly compare our approach with these contemporaneous efforts. In contrast to DiffAttack, which places a primary emphasis on creating adversarial attacks that are both imperceptible and transferable, Diff-PGD (Xue et al., 2023) prioritizes controllability and stealthiness. Diff-PGD combines PGD (Madry et al., 2018) with diffusion models, exploring its applicability across various attack types, including style-guided and physical attacks. ACA (Chen et al., 2023), on the other hand, closely aligns its research focus and techniques with those of DiffAttack. The key distinctions lie in ACA's integration of the momentum concept (Dong et al., 2018) to enhance transferability, as well as the design of a differentiable boundary process aimed at preventing boundary leakage. In contrast, our approach introduces $\mathcal{L}_{transfer}$ and $\mathcal{L}_{structure}$ to address transferability and imperceptibility, respectively.

Furthermore, we offer insights into potential future directions for diffusion-based adversarial attacks. One avenue for future research is to take diffusion models as a novel input augmentation. Recently, there are many works (Long et al., 2022; Xie et al., 2019) that enhance the attack's transferability by applying differentiable augmentations on the input image, in which way, the crafted adversarial examples gain robustness under different scenarios. In line with these approaches, we can also take diffusion models as novel augmentations. By directly adding noise (or applying DDIM Inversion), we first convert the input image into the latent space, then we conduct the diffusion denoising process to reconstruct images. This reconstruction process, with small differences from the input image every time, can be seen as an augmentation when we leverage stochastic sampling in each step (the way like DDPM (Ho et al., 2020) but not deterministic DDIM (Song et al., 2021)). Therefore, we may expect good transferability in this way.

Moreover, as the adversarial example crafted by diffusion models has many semantic clues embedded in it (see Figure 1), it is also interesting and worth exploring whether the accuracy of clean images can be improved if we merge these examples in the training dataset and whether such an adversarial training can enhance the robustness of the classifier without sacrificing the clean image accuracy compared with previous attacks (Kurakin et al., 2017).

Additionally, we identify three crucial aspects of diffusion-based attacks that merit further examination. First, the substantial computational cost, arising from the iterative nature and numerous parameters of diffusion models, potentially limits their practicality in real-time or resource-constrained settings (see Appendix M). Second, compared to pixel-based attacks, DiffAttack struggles to achieve a 100% white-box attack success rate, a phenomenon also observed in other generative-model-based (GAN-based) attacks (Poursaeed et al., 2018) and unrestricted attacks (Zhao et al., 2020; Yuan et al., 2022) (see Table 1 and Appendix J). Finally, in the transferable targeted attack task (see Appendix L), DiffAttack, along with other compared attacks, exhibits low transferability despite strong performance in the untargeted attack task. These findings also suggest promising avenues for future research.

## 6 CONCLUSION

In this work, we explore the potential of diffusion models in crafting adversarial examples and propose a powerful transfer-based unrestricted attack. By leveraging the properties of diffusion models, our approach achieves both imperceptibility and transferability. Experiments across extensive black-box models, defenses, and datasets have demonstrated our method's superiority. Furthermore, we also comprehensively discussed the possible future work with diffusion models. We hope that our work can pave the way for further research on diffusion-based adversarial attacks.

**Ethics Statement.** Since images crafted by *DiffAttack* are natural from human eyes but can lead to wrong decisions across various black-box models and defenses, some could maliciously use our method to undermine real-world applications, inevitably raising more concerns about AI safety.

**Reproducibility Statement.** In terms of reproducibility, we have provided thorough descriptions of our method's architectures and algorithms in Section 3. For additional implementation specifics, please refer to Section 4.1 and Appendix F. Moreover, our source code, along with detailed comments and instructions, has been submitted in the *Supplemental Materials* to facilitate a thorough understanding of our work.

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

## A  OVERVIEW

In the appendix, we will first provide a detailed review of DDIM Inversion in Appendix B. Then, we conduct an analysis of the effect of perturbations on text embeddings in Appendix C. In Appendix D and Appendix E, considering the possible help for future research, we display our further trials (with little success) on improving the imperceptibility and transferability of the attacks. In Appendix F, we present more implementation details of the compared methods in Table 1 of the main paper. We compare DiffAttack with the combination of multiple attack approaches in Appendix G. We present DiffAttack's performance on more surrogate models in Appendix H. Comparisons on more datasets and with GAN-based attacks are presented in Appendix I and Appendix J respectively. In Appendix K, we present comparisons with ensemble attacks and also give a detailed analysis of the relationship between DiffAttack and the ensemble attacks. Besides, we give discussions about the DiffAttack's performance on the transferable targeted attack in Appendix L. Limitations of computational cost are discussed in Appendix M. Finally, more ablation studies, quantitive studies, and visualizations are shown in Appendix N and Appendix O.

## B  DETAILED FORMULATION OF DDIM INVERSION

Denoising Diffusion Probabilistic Models (DDPMs) (Ho et al., 2020) are a class of generative models that sample images by gradually denoising an initial Gaussian noise. There is a *forward process* and a *reversed process* in DDPMs. The *forward process* is to gradually add Gaussian noise to the original image $x_0$ and thus produces a series of noisy latents $x_1, x_2, \cdots, x_T$:

$$q(x_t|x_{t-1}) = \mathcal{N}(\sqrt{1-\beta_t}x_{t-1}, \beta_t\mathbf{I}) \tag{7}$$

where $\beta_t \in (0, 1)$. When $T$ is large enough, the last latent $x_T$ will approximately follow an isotropic Gaussian distribution.

Instead of iteratively calculating the intermediate latents to get $x_t$, a good property of the *forward process* is that we can directly sample $x_t$ from $x_0$:

$$q(x_t|x_0) = \mathcal{N}(\sqrt{\bar{\alpha}_t}x_0, (1-\bar{\alpha}_t)\mathbf{I}) \tag{8}$$

$$x_t = \sqrt{\bar{\alpha}_t}x_0 + \sqrt{1-\bar{\alpha}_t}\epsilon, \ \ \epsilon \sim \mathcal{N}(0, \mathbf{I}) \tag{9}$$

where $\alpha_t = 1 - \beta_t$, $\bar{\alpha}_t = \prod_{s=0}^{t}\alpha_s$.

The *reversed process* is to draw a new sample from the distribution $q(x_0)$. Starting from $x_T \sim \mathcal{N}(0, \mathbf{I})$, we can get a new sample by iteratively sampling the posteriors $q(x_{t-1}|x_t)$. Since $q(x_{t-1}|x_t)$ is intractable due to the unknown data distribution $q(x_0)$, a neural network $p_\theta$ is trained to approximate that by predicting the mean and covariance of $q(x_{t-1}|x_t)$, which is shown to also be Gaussian distributions (Sohl-Dickstein et al., 2015):

$$p_\theta(x_{t-1}|x_t) = \mathcal{N}(\mu_\theta(x_t, t), \Sigma_\theta(x_t, t)) \tag{10}$$

Since $\mu_\theta(x_t, t) = \frac{1}{\sqrt{\alpha_t}}\left(x_t - \frac{\beta_t}{\sqrt{1-\bar{\alpha}_t}}\epsilon_\theta(x_t, t)\right)$, Ho *et al.* (Ho et al., 2020) simplified the objective function by only predicting the noise $\epsilon_\theta(x_t, t)$:

$$\min_\theta \mathcal{L}(\theta) = \mathbb{E}_{x_0, \epsilon \sim N(0,I), t}\|\epsilon - \epsilon_\theta(x_t, t)\|_2^2 \tag{11}$$

After we get the trained $\epsilon_\theta(x_t, t)$, we can conduct a sampling as follows:

$$x_{t-1} = \mu_\theta(x_t, t) + \sigma_t z, \ \ z \sim N(0, I). \tag{12}$$

Since the classic DDPMs are essentially a Markov chain and they require a large timestep $T$ to achieve good performance. To accelerate DDPMs sampling process, Song *et al.* (Song et al., 2021) generalize DDPMs from a particular Markovian process to non-Markovian processes:

$$x_{t-1} = \sqrt{\bar{\alpha}_{t-1}}\left(\frac{x_t - \sqrt{1-\bar{\alpha}_t}\epsilon_\theta(x_t)}{\sqrt{\bar{\alpha}_t}}\right) + \sqrt{1-\bar{\alpha}_{t-1}-\sigma_t^2} \cdot \epsilon_\theta(x_t) + \sigma_t z, \ \ z \sim N(0, I) \tag{13}$$

By setting $\sigma_t = 0$, we then get a deterministic sampling process (from $x_T$ to $x_0$), which is the DDIM's principle.

Since the deterministic process of DDIM can be further taken as Euler integration for solving ordinary differential equations (ODEs)(Song et al., 2021), we can map a real image back to its corresponding latent by reversing the process. This operation, named DDIM Inversion, paves the way for later editing of real images (Couairon et al., 2022; Mokady et al., 2022). By rewriting Eq. 13, the denoising process of DDIM is as follows:

$$x_{t-1} - x_t = \sqrt{\bar{\alpha}_{t-1}} \left[ \left( \sqrt{1/\bar{\alpha}_t} - \sqrt{1/\bar{\alpha}_{t-1}} \right) x_t + \left( \sqrt{1/\bar{\alpha}_{t-1} - 1} - \sqrt{1/\bar{\alpha}_t - 1} \right) \epsilon_\theta(x_t) \right] \quad (14)$$

We can then encode the real image into the latent space by reversing the above formulation:

$$x_{t+1} - x_t = \sqrt{\bar{\alpha}_{t+1}} \left[ \left( \sqrt{1/\bar{\alpha}_t} - \sqrt{1/\bar{\alpha}_{t+1}} \right) x_t + \left( \sqrt{1/\bar{\alpha}_{t+1} - 1} - \sqrt{1/\bar{\alpha}_t - 1} \right) \epsilon_\theta(x_t) \right] \quad (15)$$

## C  PERTURBATION ON GUIDED TEXT EMBEDDINGS

As mentioned in Section 3.2 in the main paper, we choose to perturb the latent $x_t$ but not the guided text $C$, which is different from the mainstream image editing approaches (Couairon et al., 2022; Mokady et al., 2022; Parmar et al., 2023). The reason is that text perturbation will be hard to transfer to other black-box models. In the following, we display the details of text perturbation designs and some necessary experiments and analyses.

### C.1  DESIGN DETAILS

Here we first define two text prompts: $C_1, C_2$, which are the first and second most possible categories predicted by the classifier. We leverage $C_1$ for the optimization of unconditional embeddings mentioned in Section 3.4 in the main paper. Then, we replace $C_1$ with $C_2$ which follows Mokady et al. (2022); Hertz et al. (2022) and can expect the changes of object semantics in the image. For the loss functions, we remove $\mathcal{L}_{transfer}$ in Eq. 6 in the main paper, and modify $\mathcal{L}_{attack}$ as follows:

$$\arg\min_{C_2} \mathcal{L}_{attack} = J(x', C_2; G_\phi) \quad (16)$$

The equation above is similar to the objective function of targeted attacks, and the insight is to trick the classifier into predicting the nearest wrong label. Other implementation details are the same as Section 4.1 in the main paper.

### C.2  EXPERIMENTS AND ANALYSIS

In this subsection, we compare the results between the text perturbation and the latent perturbation. From Table 5, we can observe that although the text perturbation has a slightly higher attack success in a white-box way (0.5 point accuracy lower on Inc-v3), the attack itself is hard to work on the other black-box models, thus not competitive with the latent perturbations. We attribute this phenomenon to the fact that the text perturbation is more high-level than the latent perturbation, due to text semantics. Therefore, it will tend to generate more realistic results (lower FID in Table 5), but has limited control over the local area, while the latent perturbation does the opposite.

Table 5: **Comparisons of perturbations on the latent and text.** "S." denotes surrogate models while "T." denotes target models. For the white-box attacks (surrogate model same as target one), we set their background to gray. "AVG(w/o self)" denotes the average accuracy on all the target models except the one that same as the surrogate one. The best result is bolded.

| T.
S. | CNNs | | | | | Transformers | | | | MLPs | | AVG(w/o self) | FID |
|---|---|---|---|---|---|---|---|---|---|---|---|---|---|
| | Res-50 | VGG-19 | Mob-v2 | Inc-v3 | ConvNeXt | ViT-B | Swin-B | DeiT-B | DeiT-S | Mix-B | Mix-L | | |
| Clean | 92.7 | 88.7 | 86.9 | 80.5 | 97.0 | 93.7 | 95.9 | 94.5 | 94.0 | 82.5 | 76.5 | 89.4 | 57.8 |
| Text Perturbation | 79.6 | 73.3 | 74.7 | **13.4** | 91.3 | 85.9 | 86.9 | 87.9 | 86.0 | 71.6 | 63.8 | 80.1 | **58.8** |
| Latent Perturbation | **59.5** | **55.6** | **55.4** | 13.9 | **76.9** | **75.2** | **72.8** | **74.0** | **71.0** | **58.9** | **54.7** | **65.4** | 62.3 |

## D  TRIAL FOR BETTER IMPERCEPTIBILITY WITH "PSEUDO" MASK

As mentioned in Section 3.4, for some specific images, the adversarial examples crafted by *DiffAttack* may distort a lot compared with the original ones. For better control of the changes, we try to generate

Table 6: **Comparisons of different mask types and upsampling strategies.** For the white-box attacks (surrogate model same as target one), we set their background to gray. "AVG(w/o self)" denotes the average accuracy on all the target models except the one that same as the surrogate one. The best result is bolded.

| Mask Types | Upsampling Strategy | CNNs | | | | | Transformers | | | | MLPs | | AVG(w/o self) | FID |
|---|---|---|---|---|---|---|---|---|---|---|---|---|---|---|
| | | Res-50 | VGG-19 | Mob-v2 | Inc-v3 | ConvNeXt | ViT-B | Swin-B | DeiT-B | DeiT-S | Mix-B | Mix-L | | |
| None | None | **59.5** | **55.6** | **55.4** | 13.9 | 76.9 | 75.2 | 72.8 | **74.0** | 71.0 | 58.9 | 54.7 | 65.4 | 62.3 |
| hard | nearest | 71.4 | 67.9 | 64.8 | 17.8 | 85.2 | 80.9 | 82.9 | 80.3 | 81.0 | 68.2 | 60.2 | 74.2 | 59.1 |
| hard | bilinear | 68.8 | 66.8 | 65.6 | 18.3 | 84.0 | 79.0 | 81.4 | 79.9 | 79.5 | 66.0 | 61.8 | 73.3 | 59.3 |
| soft | bilinear | 73.9 | 69.4 | 66.9 | 18.4 | 88.2 | 82.7 | 86.5 | 84.8 | 82.0 | 68.5 | 62.0 | 76.5 | **58.8** |

"pseudo" masks with the cross attention. With these masks, we can then filter out the background regions and only perturb the foreground objects, thus achieving better human-imperception. However, we found that although the results could more easily evade the human eyes, their transferability dropped a lot. We infer this may be because background information is also beneficial for image recognition. More details about the implementation and experiments of the trial can be found as follows. In practice, we will weaken the inversion strength for overly distorted images.

## D.1 DESIGN DETAILS

As mentioned in Section 3.3 in the main paper, there is a strong relationship in the cross-attention maps between the text prompt and the image pixels. Thus, we can make use of this property to generate the true label's "pseudo" mask:

$$P = \text{Average}(\text{Cross}(x_t, t, C; \text{SDM}))$$

(17)

$$M_{soft} = \text{Up}(\frac{P}{\text{Max}(P)})$$

(18)

$$(\text{Optional}) \quad M_{hard} = \begin{cases} 1, & M_{soft} > 0.5 \\ 0, & M_{soft} \leq 0.5 \end{cases}$$

(19)

where $\text{Up}(\cdot)$ is an upsampling operation to resize the cross-attention map (due to the existing downsamplings in the encoder of the Autoencoder and U-Net). $\text{Max}(\cdot)$ is to extract the maximum value and normalize the cross-attention maps $P$. Since $P \geq 0$, the normalized $M_{soft} \in [0, 1]$. Eq. 19 is optional to get a hard mask. With the mask, we then filter out the background area and only apply perturbations on the foreground (area covered by true objects). The Eq. 3 in the main paper is then changed as follows:

$$\arg\min_{x_t} \mathcal{L}_{attack} = -J(x' \times M + x \times (1 - M), y; G_\phi)$$

(20)

The optimization details are the same as the implementation details in Section 4.1 in the main paper.

## D.2 EXPERIMENTS AND ANALYSIS

Here we conduct experiments to see the impact of different upsampling strategies and different mask types. In Table 6, we display the performance when the mask is applied. It can be perceived from the results that there is an obvious trade-off between transferability and imperceptibility. The use of masks lowers the FID value, yet also lowers the attack success by a large margin. We infer that it is because the recognition of an image is not only related to its foreground but also its background (Zhu et al., 2017). Thus the attack success rate will drop when the mask is applied. We also visualize the adversarial example crafted by leveraging the mask in Figure 6, from which we can see that the applied mask can better preserve words on hot air balloon skin, and the hard mask tends to generate blocky artifacts compared with soft-mask.

## E TRIAL ON FURTHER IMPROVING TRANSFERABILITY

We also explore further improving the transferability of *DiffAttack*. For image classification, the classifier will output each category's confidence, and *top1* is usually taken as the final decision. We here try to also make use of the following 4 categories in *top5* for better transferability. Specifically,

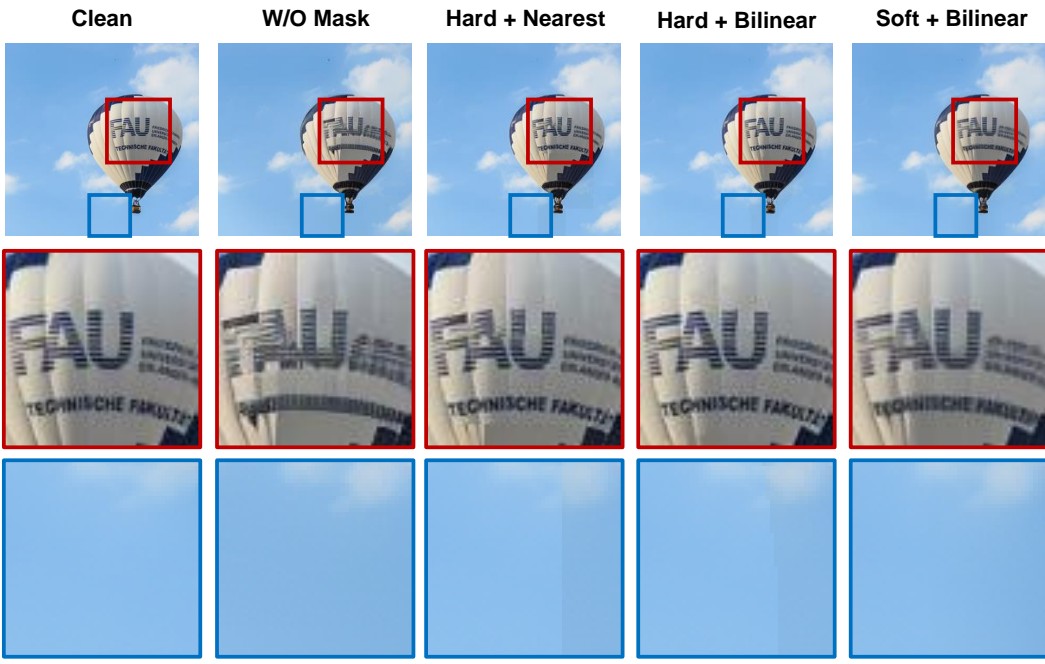

Figure 6: **Visualization of the adversarial example crafted by leveraging the mask.** The second and third rows denote the scaled-up regions in the first row.

different from Section 3.3 where we only set the guided text to *top1* category's name, we here set the text to a stack of the 5 categories' names in *top5* (sort by confidence from largest to smallest). Then, we optimize $x_t$ to reduce the intensity of cross attention between pixels and the first category text and increase that between pixels and the other four categories text. The motivation is that, the confidence denotes, to some extent, the amount of related information of the category in the image, thus it may be much easier to deceive the classifier to the nearest category on the decision plane. However, from our experiments, this trial fails to improve the transferability and even hurts it. We attribute this to the limitation of the search space. More details can be found as follows.

### E.1 DESIGN DETAILS

In Eq. 4 in the main paper, $C =$ "{True Label / Category $1_{\text{st}}$}" that the guided text can be either the true label or the *top1* predicted category. We here extend the text to leverage more categories:

$$C_{ext} = \text{"\{Category } 1_{\text{st}}\}, \text{\{Category } 2_{\text{nd}}\}, \cdots, \text{\{Category N}_{\text{th}}\}\text{"} \tag{21}$$

where {Category $N_{\text{th}}$} denotes the name of the $N_{th}$ most possible category predicted by the classifier. Then, Eq. 3 in the main paper is modified to:

$$\mathcal{L}_{attack} = -J(x', \text{Category } 1_{\text{st}}; G_\phi) + \underbrace{(J(x', \text{Category } 2_{\text{nd}}; G_\phi) + \cdots + J(x', \text{Category N}_{\text{th}}; G_\phi))}_{N-1} \tag{22}$$

By minimizing the above equation, the adversarial examples are crafted to lead the classification results towards the most error-prone categories, which may have benefits on the transferability (but failed through our experiments). We further add an extra loss to force the perturbed $x_t$ to have lower cross-attention intensity with {Category $1_{\text{st}}$} and higher with other text prompts, which we expect can help deceive the diffusion models:

$$P_i = \text{Average}(\text{Cross}(x_t, t, C_i; \text{SDM})) \tag{23}$$

$$\arg\max_{x_t} \mathcal{L}_{ext} = \text{Average}(\underbrace{P_2 + \cdots + P_N}_{N-1}) - P_1 \tag{24}$$

where $C_i$ denotes Category $i_{\text{th}}$, and $P_i$ denotes the cross attention between image pixels and $C_i$. Average($\cdot$) here represents the averaging operation in pixel space. We then add $\mathcal{L}_{ext}$ to Eq. 6 in the main paper with a weight factor set to 100.

Table 7: **Explorations on the effect of different categories as text prompts.** For the white-box attacks (surrogate model same as target one), we set their background to gray. "AVG(w/o self)" denotes the average accuracy on all the target models except the one that same as the surrogate one. The best result is bolded.

| top-N | CNNs | | | | | Transformers | | | | MLPs | | AVG(w/o self) | FID |
|---|---|---|---|---|---|---|---|---|---|---|---|---|---|
| | Res-50 | VGG-19 | Mob-v2 | Inc-v3 | ConvNeXt | ViT-B | Swin-B | DeiT-B | DeiT-S | Mix-B | Mix-L | | |
| 1 | **59.5** | **55.6** | **55.4** | 13.9 | **76.9** | **75.2** | **72.8** | **74.0** | **71.0** | **58.9** | 54.7 | **65.4** | 62.3 |
| 2 | 67.0 | 63.7 | 61.2 | 9.1 | 81.7 | 77.1 | 77.5 | 79.1 | 76.4 | 64.0 | 56.8 | 70.4 | **60.9** |
| 5 | 64.4 | 61.0 | 59.2 | **5.9** | 81.4 | 78.8 | 78.0 | 77.5 | 75.9 | 61.0 | **54.0** | 69.1 | 62.8 |

Table 8: **Comparison with the combination of multiple attack approaches.** We report top-1 accuracy(%) of each method. For the white-box attacks (surrogate model same as target one), we set their background to gray. "AVG(w/o self)" denotes the average accuracy on all the target models except the one that same as the surrogate one. The best result is bolded.

| Attacks | CNNs | | | | | Transformers | | | | MLPs | | AVG(w/o self) | FID |
|---|---|---|---|---|---|---|---|---|---|---|---|---|---|
| | Res-50 | VGG-19 | Mob-v2 | Inc-v3 | ConvNeXt | ViT-B | Swin-B | DeiT-B | DeiT-S | Mix-B | Mix-L | | |
| S$^2$I-FGSM | 17.9 | 0 | 11.3 | 31.8 | 49.5 | 74.1 | 57.9 | 76.0 | 68.0 | 52.6 | 50.8 | 49.0 | 82.9 |
| S$^2$I-MI-FGSM | 6.2 | 0 | 3.6 | 14.5 | 30.1 | 51.4 | 41.1 | 54.3 | 45.7 | 34.5 | 33.0 | 31.4 | 100.0 |
| S$^2$I-DI-MI-FGSM | 3.6 | 0 | 2.3 | 9.2 | 24.6 | 44.7 | 33.3 | 49.2 | 38.2 | 28.2 | 29.1 | 26.2 | 104.5 |
| S$^2$I-TI-DI-MI-FGSM | 5.2 | 0 | 3.1 | 7.8 | 40.0 | 35.9 | 46.0 | 49.3 | 36.8 | 27.5 | 27.1 | 27.9 | 104.9 |
| S$^2$I-SI-TI-DI-MI-FGSM | 5.5 | 0 | 4.1 | 7.7 | 45.4 | 34.4 | 47.5 | 49.5 | 36.4 | 27.0 | 26.3 | 28.4 | 114.7 |
| DiffAttack(Ours) | 21.1 | 2.7 | 19.4 | 29.7 | 43.1 | 52.9 | 41.6 | 51.3 | 45.0 | 39.6 | 38.5 | 38.2 | **63.9** |
| DiffAttack(w/o Structure Controls) | 19.7 | 3.9 | 15.5 | 19.9 | 32.2 | 35.0 | 28.8 | 30.8 | 30.1 | 20.7 | 21.8 | **25.5** | 96.2 |

## E.2 EXPERIMENTS AND ANALYSIS

We here analyze the impact of different numbers of categories leveraged as text prompts. From Table 7, leveraging more guided category texts failed to improve the attack's transferability, and even damage the performance. We infer that it is because the search space of the attack is limited when we force the adversarial examples to be classified as some specific categories. When we set the category number from 2 to 5, we can observe a slight increase in the attack success, while when we set it to 1, we have no constraint on the predicted category, and thus gain a large increase in the attack success.

## F MORE IMPLEMENTATION DETAILS

In Section 4 in the main paper, we compared the performance of *DiffAttack* with other transfer-based black-box attacks, including MI-FGSM (Dong et al., 2018), DI-FGSM (Xie et al., 2019), TI-FGSM (Dong et al., 2019), PI-FGSM (Gao et al., 2020), S$^2$I-FGSM (Long et al., 2022), and also two unrestricted attacks (PerC-AL (Zhao et al., 2020), and NCF (Yuan et al., 2022)). Here, we detail the hyperparameter settings of the compared attack methods.

The settings mostly follow their source papers/codes. All I-FGSM-based ones (Dong et al., 2018; Xie et al., 2019; Dong et al., 2019; Gao et al., 2020; Long et al., 2022) are constrained by $L_{inf}$ with steps set to 10, maximum perturbation set to 16, and step size set to 1.6. For MI-FGSM, we set its decay factor to 1.0. For DI-FGSM, we set its transformation probability to 0.5. For TI-FGSM, we set its kernel size to 7. For PI-FGSM, we set its amplification factor to 10. For S$^2$I-FGSM, we set its inner iteration number to 20, its tuning factor to 0.5, and its standard deviation to 16. As for the unrestricted attacks (Zhao et al., 2020; Yuan et al., 2022), we set PerC-AL's iteration number to 1000 and its confidence to 40. For NCF, we set its random search number to 50, neighborhood search number to 15, reset number to 10, and step size to 0.013.

## G COMPARISON WITH A COMBINATION OF MULTIPLE ATTACK APPROACHES

Many recent $L_p$-norm-based attacks enhance their efficacy by combining with other attack strategies. For instance, the S$^2$I-SI-TI-DIM (Long et al., 2022) approach integrates five attack methods (MI-FGSM (Dong et al., 2018), DI-FGSM(Xie et al., 2019), TI-FGSM(Dong et al., 2019), SI-FGSM(Lin et al., 2020), and their own S$^2$I-FGSM). While it is unfair to compare a single DiffAttack against an ensemble of these attack strategies, we still perform such comparisons in Table 8 to better elucidate

Table 9: **Comparisons on more surrogate models.** We report top-1 accuracy(%) of each method. "S." denotes surrogate models while "T." denotes target models. For the white-box attacks (surrogate model same as target one), we set their background to gray. "AVG(w/o self)" denotes the average accuracy on all the target models except the one that same as the surrogate one. The best result is bolded, and the second-best result is underlined.

| S. \ T. | Attacks | CNNs | | | | | Transformers | | | | MLPs | | AVG(w/o self) | FID |
|---|---|---|---|---|---|---|---|---|---|---|---|---|---|---|
| | | Res-50 | VGG-19 | Mob-v2 | Inc-v3 | ConvNeXt | ViT-B | Swin-B | DeiT-B | DeiT-S | Mix-B | Mix-L | | |
| | Clean | 92.7 | 88.7 | 86.9 | 80.5 | 97.0 | 93.7 | 95.9 | 94.5 | 94.0 | 82.5 | 76.5 | 89.4 | 57.8 |
| ViT-B | PI-FGSM | 34.2 | 27.7 | 23.6 | 31.5 | 66.9 | 0 | 56.5 | 25.6 | 17.0 | 29.7 | 26.3 | 33.9 | 91.2 |
| | S²I-FGSM | 45.0 | 39.6 | 38.6 | 38.1 | 63.1 | 0.2 | 45.2 | 10.7 | 5.5 | 18.1 | 20.2 | 32.4 | 70.2 |
| | NCF | 45.1 | 40.4 | 39.6 | 56.1 | 73.5 | 27.6 | 70.1 | 64.1 | 57.8 | 49.7 | 44.9 | 54.1 | 67.4 |
| | DiffAttack(Ours) | 39.4 | 40.5 | 36.1 | 34.7 | 41.7 | 4.7 | 30.3 | 22.4 | 19.9 | 27.2 | 30.0 | **32.2** | **66.4** |
| DeiT-B | PI-FGSM | 33.8 | 19.4 | 22.8 | 30.6 | 64.7 | 22.5 | 54.5 | 0 | 16.7 | 32.6 | 28.9 | 34.4 | 92.1 |
| | S²I-FGSM | 39.8 | 34.5 | 29.3 | 32.6 | 50.4 | 6.7 | 28.0 | 0.4 | 3.9 | 13.6 | 17.9 | **25.7** | 75.8 |
| | NCF | 52.2 | 47.3 | 46.7 | 59.3 | 73.4 | 62.5 | 67.5 | 31.7 | 59.2 | 50.9 | 48.0 | 56.7 | **65.3** |
| | DiffAttack(Ours) | 39.9 | 40.4 | 36.8 | 37.0 | 37.5 | 22.4 | 25.9 | 3.1 | 18.2 | 26.2 | 27.9 | 31.2 | 67.6 |
| Mix-B | PI-FGSM | 47.5 | 37.0 | 39.2 | 40.8 | 72.1 | 49.9 | 70.9 | 56.6 | 45.1 | 0 | 13.9 | 47.3 | 85.5 |
| | S²I-FGSM | 60.6 | 52.4 | 47.8 | 52.1 | 72.6 | 48.4 | 58.4 | 43.9 | 40.7 | 1.6 | 8.9 | 48.6 | 66.4 |
| | NCF | 55.0 | 47.5 | 49.6 | 61.1 | 81.8 | 71.1 | 77.5 | 75.6 | 71.1 | 10.0 | 35.0 | 62.5 | 65.2 |
| | DiffAttack(Ours) | 52.2 | 52.1 | 49.6 | 45.0 | 57.9 | 48.8 | 49.9 | 44.6 | 45.4 | 16.6 | 22.1 | **46.8** | **64.2** |

Table 10: **Imperceptibility assessment under LPIPS metric.** We report the LPIPS value of different attack methods under different surrogate models. "S." denotes surrogate models while "A." denotes attack methods. The best result is bolded.

| S. \ A. | PI-FGSM | S²I-FGSM | NCF | DiffAttack(Ours) |
|---|---|---|---|---|
| Res-50 | 0.356 | 0.157 | 0.383 | **0.137** |
| VGG-19 | 0.367 | 0.155 | 0.392 | **0.150** |
| Mob-v2 | 0.367 | 0.157 | 0.387 | **0.138** |
| Inc-v3 | 0.368 | 0.137 | 0.343 | **0.126** |
| ConvNeXt | 0.359 | 0.159 | 0.360 | **0.154** |
| Swin-B | 0.358 | **0.114** | 0.346 | 0.138 |
| ViT-B | 0.360 | 0.177 | 0.364 | **0.152** |
| DeiT-B | 0.362 | 0.166 | 0.336 | **0.146** |
| Mix-B | 0.344 | 0.154 | 0.326 | **0.143** |

the capabilities of DiffAttack. The adversarial examples are crafted on VGG-19, with the powerful S²I-SI-TI-DIM attack serving as the reference.

The results indicate that S²I-based methods exhibit improved transferability when combined with other attacks, albeit at the cost of increased distortion. Our original DiffAttack cannot surpass the performance achieved by the combination of multiple attack approaches. Nevertheless, when structural controls are eliminated (as discussed in Section 3.4) to align the FID values for fair comparisons, DiffAttack once again showcases superior performance.

# H PERFORMANCE ON ADDITIONAL SURROGATE MODELS AND LPIPS METRIC

Besides the results in Table 1 in the main paper, we supplement more experiments when the surrogate models are Transformers or MLPs in Table 9. Here, we further consider ViT-B, DeiT-B, and Mix-B as the surrogate model. For brevity, we only compare DiffAttack with those more recent attack methods (Gao et al., 2020; Long et al., 2022; Yuan et al., 2022). From the results, it is further verified that DiffAttack generalizes well on various model structures, achieving good performance on both imperceptibility and transferability.

Furthermore, to bolster the credibility of our imperceptibility assessment in the main paper, we augment our evaluation beyond FID by incorporating the full-reference image quality assessment metric, LPIPS (Zhang et al., 2018). The outcomes in Table 10 reveal that DiffAttack also excels in terms of LPIPS, further solidifying its efficacy for achieving superior imperceptibility.

Table 11: **Comparisons on CUB-200-2011 dataset and Stanford Cars dataset.** We report top-1 accuracy(%) of each method. "S." denotes surrogate models while "T." denotes target models. For the white-box attacks (surrogate model same as target one), we set their background to gray. "AVG(w/o self)" denotes the average accuracy on all the target models except the one that same as the surrogate one. The best result is bolded, and the second-best result is underlined.

| S. \ T. | Attacks | CUB-200-2011 | | | | | | Stanford Cars | | | | | |
|---|---|---|---|---|---|---|---|---|---|---|---|---|---|
| | | Res-50 | SENet154 | SE-Res101 | AVG(w/o self) | FID | LPIPS | Res-50 | SENet154 | SE-Res101 | AVG(w/o self) | FID | LPIPS |
| | clean | 75.7 | 80.5 | 76.6 | 77.6 | 11.1 | - | 73.9 | 76.4 | 74.4 | 74.9 | 11.6 | - |
| Res-50 | MI-FGSM | 3.1 | 40.7 | 32.7 | 36.7 | 31.7 | 0.340 | 0.1 | 33.5 | 25.9 | 29.7 | 41.3 | 0.251 |
| | DI-FGSM | 0.3 | 42.7 | 33.8 | 38.3 | 20.9 | 0.155 | 0.1 | 33.3 | 29.3 | 31.3 | 28.7 | **0.097** |
| | TI-FGSM | 2.8 | 50.6 | 43.9 | 47.3 | 21.1 | 0.136 | 0.1 | 46.9 | 41.0 | 44.0 | 23.2 | **0.097** |
| | PI-FGSM | 9.1 | 35.2 | 26.2 | 30.7 | 34.8 | 0.355 | 1.5 | 31.5 | 23.2 | 27.4 | 53.2 | 0.310 |
| | S²I-FGSM | 0.7 | 35.1 | 28.1 | 31.6 | 24.3 | 0.196 | 0.1 | 25.7 | 24.4 | 25.1 | 34.4 | 0.134 |
| | NCF | 0.2 | 22.7 | 13.9 | 18.3 | 35.2 | 0.335 | 6.6 | 46.0 | 38.4 | 42.2 | 24.1 | 0.302 |
| | DiffAttack(Ours) | 3.3 | 19.3 | 16.7 | **18.0** | 20.6 | **0.122** | 0.1 | 15.1 | 13.1 | **14.1** | **17.8** | 0.112 |
| SENet154 | MI-FGSM | 41.7 | 0.2 | 42.3 | 42.0 | 37.9 | 0.402 | 32.4 | 0.0 | 33.8 | 33.1 | 41.3 | 0.295 |
| | DI-FGSM | 54.5 | 0.2 | 48.9 | 51.7 | 23.5 | 0.158 | 45.6 | 0.1 | 45.5 | 45.6 | 29.1 | 0.096 |
| | TI-FGSM | 60.1 | 0.3 | 56.2 | 58.1 | 20.8 | 0.137 | 54.1 | 0.1 | 53.2 | 53.7 | 23.0 | **0.095** |
| | PI-FGSM | 30.5 | 0.0 | 33.1 | 31.8 | 46.5 | 0.403 | 27.7 | 0.0 | 26.3 | 24.1 | 59.6 | 0.333 |
| | S²I-FGSM | 43.2 | 0.0 | 34.0 | 38.6 | 25.4 | 0.164 | 27.7 | 0.0 | 25.7 | 26.7 | 33.6 | 0.108 |
| | NCF | 13.5 | 6.8 | 17.6 | **15.5** | 35.0 | 0.314 | 38.5 | 20.7 | 41.6 | 40.1 | 23.3 | 0.279 |
| | DiffAttack(Ours) | 53.8 | 2.5 | 51.3 | 52.6 | **17.9** | **0.104** | 37.3 | 0.9 | 32.5 | 34.9 | **16.2** | **0.095** |
| SE-Res101 | MI-FGSM | 32.8 | 36.0 | 0.1 | 34.4 | 41.0 | 0.395 | 25.5 | 27.6 | 0.0 | 26.6 | 44.6 | 0.285 |
| | DI-FGSM | 39.4 | 38.0 | 0.2 | 38.7 | 23.5 | 0.165 | 28.1 | 29.3 | 0.2 | 28.7 | 28.5 | 0.106 |
| | TI-FGSM | 53.4 | 55.3 | 0.2 | 54.4 | **21.8** | 0.136 | 48.7 | 49.8 | 0.0 | 49.3 | 22.5 | **0.096** |
| | PI-FGSM | 21.7 | 29.8 | 0.0 | 25.8 | 45.5 | 0.403 | 18.5 | 29.3 | 0.0 | 23.9 | 59.9 | 0.331 |
| | S²I-FGSM | 30.4 | 31.5 | 0.0 | 31.0 | 26.7 | 0.195 | 20.5 | 17.1 | 0.1 | 18.8 | 36.9 | 0.142 |
| | NCF | 9.4 | 20.2 | 3.1 | **14.8** | 33.3 | 0.316 | 33.4 | 46.8 | 12.1 | 40.1 | 24.0 | 0.298 |
| | DiffAttack(Ours) | 27.0 | 23.5 | 3.9 | 25.3 | 22.4 | **0.121** | 17.5 | 16.0 | 0.3 | **16.8** | **18.0** | 0.114 |

## I    Performance on More Datasets

In Section 4 of the main paper, our comparative experiments are exclusively conducted on the ImageNet-Compatible Dataset. To bolster the credibility of DiffAttack's performance and its applicability, we have expanded our evaluation to encompass two additional datasets: CUB-200-2011 (Wah et al., 2011) and Stanford Cars (Krause et al., 2013). Aligning with the ImageNet-Compatible dataset, we randomly selected 1,000 samples from both the CUB-200-2011 and Stanford Cars datasets, respectively, for crafting adversarial examples. For normally trained models, we employed three models: ResNet50 (R-50) (He et al., 2016), SENet154 (S-154), and SE-ResNet101 (SR-101) (Hu et al., 2018), all initialized with pretrained weights provided by Zhang et al. (2022a).

The results in Table 11 highlight DiffAttack's strong generalization across diverse datasets. Note that PerC-AL is not included in the comparison due to its notably low transferability, as indicated in Table 1 in the main paper.

## J    Comparisons with GAN-Based Attack Methods

Different from iterative optimization attacks, GAN-based attack methods craft adversarial examples by directly training a generator and thus achieve better efficiency. Considering both the GAN-based attacks and our DiffAttack leverage generative models (although DiffAttack is essentially an iterative optimization approach), we supplement comparative experiments between them to increase the comprehensiveness of our experiments and further highlight DiffAttack's benefits.

Here, we consider four GAN-based attacks: GAP (Poursaeed et al., 2018), CDA (Naseer et al., 2019), BIA (Zhang et al., 2022a), and TSAA (He et al., 2022). All these compared methods have their code open-source and our experiments are based on that. For BIA (Zhang et al., 2022a), we directly use their provided pretrained generator (for VGG-19) to generate adversarial examples. The variants of it (BIA+DA and BIA+RN) have also been considered for comparisons. For CDA (Naseer et al., 2019), we utilize their pretrained generator (for VGG-19) to generate adversarial examples. We also take recent TSAA (He et al., 2022) into account. Considering the original TSAA is a sparse attack, we directly remove its last layer's mask mechanism to allow it to attack the whole image. As TSAA does not provide pretrained weight for VGG-19 but provides for Res-50, we compare DiffAttack with it on Res-50. For GAP (Poursaeed et al., 2018), since it does not provide any pretrained weight, we strictly follow their provided training code and train the generator for VGG-19 and Res-50 ourselves. As GAP has two kinds of generator (universal and image dependent), we trained a total of four

Table 12: **Comparisons with GAN-based attack methods.** We report top-1 accuracy(%) of each method. We craft adversarial examples either on VGG-19 or Res-50. For the white-box attacks (surrogate model same as target one), we set their background to gray. "AVG(w/o self)" denotes the average accuracy on all the target models except the one that same as the surrogate one. The best result is bolded, and the second-best result is underlined.

| Attacks | CNNs | | | | | Transformers | | | | MLPs | | AVG (w/o self) | FID | LPIPS |
|---|---|---|---|---|---|---|---|---|---|---|---|---|---|---|
| | Res-50 | VGG-19 | Mob-v2 | Inc-v3 | ConvNeXt | ViT-B | Swin-B | DeiT-B | DeiT-S | Mix-B | Mix-L | | | |
| clean | 92.7 | 88.7 | 86.9 | 80.5 | 97 | 93.7 | 95.9 | 94.5 | 94 | 82.5 | 76.5 | 89.4 | 57.8 | - |
| GAP (universal) | 56.9 | 12.4 | 20.6 | 56.9 | 92.2 | 92.1 | 91.3 | 91.1 | 88.0 | 65.1 | 57.5 | 71.2 | 100.6 | 0.178 |
| GAP(image dependent) | 70.1 | 9.5 | 35.6 | 60.2 | 79.4 | 89.6 | 89.1 | 88.6 | 83.6 | 66.1 | 55.2 | 71.8 | 108.0 | 0.164 |
| CDA | 23.0 | 0.2 | 16.5 | 48.6 | 45.2 | 89.1 | 80.7 | 86.0 | 82.4 | 62.7 | 54.1 | 58.8 | 131.8 | 0.174 |
| BIA | 25.2 | 1.8 | 10.5 | 38.6 | 58.8 | 83.2 | 75.6 | 82.9 | 80.3 | 54.3 | 47.6 | 55.7 | 200.3 | 0.252 |
| BIA+DA | 16.3 | 1.6 | 7.6 | 33.1 | 44.0 | 85.1 | 74.8 | 84.5 | 80.4 | 55.7 | 49.8 | 53.1 | 246.0 | 0.247 |
| BIA+RN | 14.7 | 1.4 | 5.8 | 28.6 | 52.2 | 79.7 | 70.0 | 80.7 | 77.4 | 48.9 | 44.1 | 50.2 | 246.7 | 0.275 |
| DiffAttack(Ours) | 21.1 | 2.7 | 19.4 | 29.7 | 43.1 | 52.9 | 41.6 | 51.3 | 45.0 | 39.6 | 38.5 | **38.2** | **63.9** | **0.150** |

| Attacks | CNNs | | | | | Transformers | | | | MLPs | | AVG (w/o self) | FID | LPIPS |
|---|---|---|---|---|---|---|---|---|---|---|---|---|---|---|
| | Res-50 | VGG-19 | Mob-v2 | Inc-v3 | ConvNeXt | ViT-B | Swin-B | DeiT-B | DeiT-S | Mix-B | Mix-L | | | |
| clean | 92.7 | 88.7 | 86.9 | 80.5 | 97 | 93.7 | 95.9 | 94.5 | 94 | 82.5 | 76.5 | 89.4 | 57.8 | - |
| GAP (universal) | 35.6 | 34.0 | 34.3 | 55.3 | 88.8 | 87.0 | 91.9 | 91.4 | 85.8 | 64.7 | 57.4 | 69.1 | 89.7 | 0.248 |
| GAP(image dependent) | 42.9 | 21.7 | 25.4 | 55.6 | 87.1 | 88.5 | 89.5 | 88.1 | 84.4 | 62.3 | 55.8 | 65.8 | 102.6 | 0.147 |
| TSAA (dense) | 15.4 | 16.4 | 22.2 | 52.0 | 74.3 | 87.4 | 89.2 | 90.4 | 86.3 | 66.6 | 61.8 | 64.7 | 105.6 | 0.261 |
| DiffAttack(Ours) | 3.7 | 24.4 | 22.9 | 31.0 | 41.0 | 48.8 | 43.8 | 49.5 | 45.0 | 42.9 | 42.2 | **39.2** | **62.6** | **0.137** |

generators. All of these methods' maximum perturbation is set to 10, which is aligned with their source paper (we also tried 16, but it will massively distort the image and lead to a quite high FID). The input resolution of these methods is 224×224×3, which also strictly follows their papers and is the same as our settings.

From the results in Table 12, it is amazing to see that our DiffAttack can surpass other GAN-based attacks by a large margin on transferability (AVG w/o self), while also keeping quite better imperceptibility (FID and LPIPS). These supplementary experiments can not only enhance the comprehensiveness of our findings but also reinforce the effectiveness of DiffAttack.

# K  DISCUSSIONS ABOUT DIFFATTACK'S RELATIONSHIP WITH ENSEMBLE ATTACKS

**DiffAttack as an "Implicit" Ensemble Attack.**   DiffAttack can be considered as an "Implicit" ensemble attack. The loss function $\mathcal{L}_{transfer}$ in Eq. 4 functions to divert the intermediate 2D cross-attention maps. This resembles the role of a zero-shot CLIP classifier (Radford et al., 2021), which aims to align the image's features with its corresponding text embedding. From this perspective, DiffAttack can be viewed as an ensemble adversarial attack, targeting both a zero-shot CLIP classifier and a surrogate classifier.

However, it's essential to highlight that, unlike explicit ensemble attacks involving multiple surrogate models behind the final output adversarial examples (Tramèr et al., 2018), DiffAttack's ensemble characteristic is "implicit". $\mathcal{L}_{transfer}$ is designed to perturb the intermediate 2D cross-attention maps of the diffusion model rather than attacking the final similarity results of an explicit CLIP classifier. This design avoids the need for an additional image classifier to generate adversarial examples, resulting in no additional memory overhead.

In summary, DiffAttack exhibits an **"implicit ensemble characteristic"** but differs significantly from typical explicit ensemble attacks.

**Comparisons with Explicit Ensemble Attacks Using a Zero-shot CLIP Classifier.**   To ensure the comprehensiveness of our experiments, we have included comparisons with ensemble attacks employing an additional explicit zero-shot CLIP classifier. Also, we adapted the original DiffAttack into an explicit ensemble attack by substituting $\mathcal{L}_{transfer}$ with an explicit CLIP surrogate model.

We display the compared results in Table 13. The base surrogate model is VGG-19 and we consider comparisons with three recent attack methods (Gao et al., 2020; Long et al., 2022; Yuan et al., 2022). For the zero-shot CLIP classifier, we utilized the pretrained ViT-B/32 weights provided by OpenAI. Based on the results obtained, our original DiffAttack consistently outperforms other methods in

Table 13: **Comparisons with explicit ensemble attacks using a zero-shot CLIP classifier.** We report top-1 accuracy(%) of each method. We craft adversarial examples on VGG-19 and CLIP. For the white-box attacks (surrogate model same as target one), we set their background to gray. "AVG(w/o self)" denotes the average accuracy on all the target models except the one that same as the surrogate one. The best result is bolded, and the second-best result is underlined.

| Attacks | CNNs | | | | | Transformers | | | | MLPs | | AVG(w/o self) | FID |
|---|---|---|---|---|---|---|---|---|---|---|---|---|---|
| | Res-50 | VGG-19 | Mob-v2 | Inc-v3 | ConvNeXt | ViT-B | Swin-B | DeiT-B | DeiT-S | Mix-B | Mix-L | | |
| PI-FGSM (VGG-19) | 22.7 | 0 | 16.4 | 29.8 | 68.3 | 68.0 | 75.7 | 79.5 | 67.6 | 50.9 | 41.8 | 52.1 | 96.4 |
| PI-FGSM (VGG-19+CLIP) | 40.2 | 21.6 | 26.2 | 33.5 | 79.5 | 57.1 | 78.6 | 71.1 | 59.7 | 49.0 | 39.2 | 53.4 | 89.5 |
| S²I-FGSM (VGG-19) | 17.9 | 0.0 | 11.3 | 31.8 | 49.5 | 74.1 | 57.9 | 76.0 | 68.0 | 52.6 | 50.8 | 49.0 | 82.9 |
| S²I-FGSM (VGG-19+CLIP) | 16.1 | 0.4 | 9.6 | 26.4 | 46.8 | 58.8 | 50.3 | 63.2 | 56.3 | 44.5 | 42.3 | 41.4 | 84.6 |
| NCF (VGG-19) | 38.3 | 6.8 | 31.5 | 52.4 | 80.5 | 67.5 | 77.6 | 77.4 | 70.6 | 53.5 | 47.2 | 59.7 | 70.4 |
| NCF (VGG-19+CLIP) | 39.9 | 9.9 | 32.0 | 53.7 | 79.3 | 66.2 | 78.5 | 77.5 | 68.4 | 54.1 | 48.4 | 59.8 | 70.4 |
| DiffAttack(VGG-19) | 21.1 | 2.7 | 19.4 | 29.7 | 43.1 | 52.9 | 41.6 | 51.3 | 45.0 | 39.6 | 38.5 | 38.2 | **63.9** |
| DiffAttack (VGG-19+CLIP, w/o $L_{transfer}$) | 27.2 | 10.0 | 24.1 | 29.4 | 44.1 | 46.1 | 41.5 | 45.1 | 39.7 | 38.7 | 36.9 | **37.3** | 64.6 |

Table 14: **Leveraging DiffAttack for Ensemble Attacks.** We report top-1 accuracy(%) of each method. We craft adversarial examples on VGG-19 and CLIP. "AVG(w/o self)" denotes the average accuracy on all the target models except the ones that have a gray background. The best result is bolded.

| Attacks | CNNs | | | | | Transformers | | | | MLPs | | AVG(w/o self) | FID |
|---|---|---|---|---|---|---|---|---|---|---|---|---|---|
| | Res-50 | VGG-19 | Mob-v2 | Inc-v3 | ConvNeXt | ViT-B | Swin-B | DeiT-B | DeiT-S | Mix-B | Mix-L | | |
| S²I-FGSM(VGG-19+Res-50) | 1.5 | 0.0 | 4.5 | 14.8 | 28.8 | 57.7 | 40.4 | 61.0 | 54.0 | 41.2 | 39.2 | 38.0 | 84.7 |
| DiffAttack(VGG-19) | 21.1 | 2.7 | 19.4 | 29.7 | 43.1 | 52.9 | 41.6 | 51.3 | 45.0 | 39.6 | 38.5 | 40.1 | 63.9 |
| DiffAttack(VGG-19+Res-50,w/o $L_{transfer}$) | 3.8 | 3.8 | 11.6 | 20.0 | 24.0 | 36.3 | 26.3 | 34.0 | 30.6 | 30.9 | 30.5 | **27.1** | **62.1** |
| S²I-FGSM(VGG-19+Swin-B) | 14.5 | 0.3 | 9.5 | 25.8 | 27.2 | 51.3 | 15.4 | 52.5 | 48.1 | 41.8 | 39.0 | 34.4 | 83.1 |
| DiffAttack(VGG-19) | 21.1 | 2.7 | 19.4 | 29.7 | 43.1 | 52.9 | 41.6 | 51.3 | 45.0 | 39.6 | 38.5 | 37.8 | **63.9** |
| DiffAttack(VGG-19+Swin-B,w/o $L_{transfer}$) | 19.3 | 6.9 | 19.4 | 26.0 | 27.6 | 33.5 | 15.6 | 30.4 | 30.1 | 30.7 | 31.4 | **27.6** | 64.2 |
| S²I-FGSM(VGG-19+Mix-L) | 17.5 | 0.3 | 12.0 | 27.9 | 43.8 | 58.0 | 46.2 | 56.1 | 51.4 | 24.6 | 10.8 | 37.5 | 83.9 |
| DiffAttack(VGG-19) | 21.1 | 2.7 | 19.4 | 29.7 | 43.1 | 52.9 | 41.6 | 51.3 | 45.0 | 39.6 | 38.5 | 38.2 | **63.9** |
| DiffAttack(VGG-19+Mix-L,w/o $L_{transfer}$) | 22.7 | 4.2 | 20.5 | 31.2 | 40.7 | 43.9 | 36.8 | 43.1 | 40.4 | 27.5 | 25.8 | **34.1** | 64.3 |
| S²I-FGSM(Res-50+ViT-B) | 1.1 | 7.7 | 5.9 | 17.5 | 39.3 | 10.9 | 40.5 | 26.7 | 20.3 | 27.2 | 29.6 | **23.9** | 79.6 |
| DiffAttack(Res-50) | 3.7 | 24.4 | 22.9 | 31.0 | 41.0 | 48.8 | 43.8 | 49.5 | 45.0 | 42.9 | 42.2 | 38.1 | **62.6** |
| DiffAttack(Res-50+ViT-B,w/o $L_{transfer}$) | 6.3 | 18.8 | 18.7 | 24.6 | 27.7 | 12.9 | 26.4 | 24.2 | 21.1 | 26.5 | 27.8 | 24.0 | 63.6 |

terms of both transferability and imperceptibility, even when those methods attack an additional CLIP classifier. As for our adapted ensemble DiffAttack, which replaces $\mathcal{L}_{transfer}$ with an explicit CLIP classifier, we observed an improvement in transferability but a reduction in imperceptibility. It's worth noting again that, unlike the explicit CLIP classifier, $\mathcal{L}_{transfer}$ utilizes intermediate cross-attention maps during the denoising process, incurring no additional memory costs.

**Leveraging DiffAttack for Ensemble Attacks.** Here, we unveil another remarkable potential of diffusion models in crafting adversarial examples: **Ensemble attacks founded on diffusion models can significantly outperform conventional ensemble attacks** (Tramèr et al., 2018).

To demonstrate this, we conducted a comparison between DiffAttack and $L_p$-norm-based attacks involving multiple surrogate models, using S²I-FGSM (Long et al., 2022) as an example. Adversarial examples were generated to target various model structures.

The results in Table 14 indicate that our original DiffAttack, which targets a single model structure, falls short when compared to ensemble attacks that target two model structures explicitly. The reason is evident: when more model structures are explicitly attacked, the generated adversarial examples exhibit superior transferability across these surrogate structures. It's important to note that the diffusion model (Rombach et al., 2022) we employ, designed initially for image synthesis, fundamentally serves as an "implicit" recognition model. Therefore, our deception loss $L_{transfer}$ cannot be designed in the same manner as commonly used attack losses (See $L_{attack}$ in Eq. 3) that directly target the classifier's decision (the ultimate goal of the attack). This limitation explains the original DiffAttack's inability to outperform ensemble attacks in terms of transferability, although it still maintains superior imperceptibility.

However, when we employed an explicit ensemble attack based on DiffAttack, while also removing $L_{transfer}$ for fairness, DiffAttack achieved better (or competitive) results in both transferability and imperceptibility, as evident in Table 14. These findings underscore the potential of diffusion models as a promising platform also for constructing ensemble attacks.

Table 15: **Performance comparisons on targeted transferable attacks.** We report **Attack Success Rate**(%) of each method here. We craft adversarial examples on VGG-19. "Success Rate AVG(w/o self)" denotes the average attack success rate on all the target models except the ones that have a gray background. The best result is bolded.

| Targeted Attacks | CNNs | | | | | Transformers | | | | MLPs | | Success Rate AVG(w/o self) | FID |
|---|---|---|---|---|---|---|---|---|---|---|---|---|---|
| | Res-50 | VGG-19 | Mob-v2 | Inc-v3 | ConvNeXt | ViT-B | Swin-B | DeiT-B | DeiT-S | Mix-B | Mix-L | | |
| MI-FGSM | 0.6 | 99.5 | 0.6 | 0.1 | 0.4 | 0.0 | 0.0 | 0.0 | 0.0 | 0.0 | 0.0 | 0.2 | 93.5 |
| DI-FGSM | 0.4 | 94.7 | 0.5 | 0.2 | **0.1** | 0.0 | 0.1 | 0.0 | 0.0 | 0.0 | 0.0 | 0.1 | 75.6 |
| TI-FGSM | 0.3 | 97.3 | 0.1 | 0.1 | 0.0 | 0.0 | 0.0 | 0.0 | 0.0 | 0.0 | 0.0 | 0.1 | 70.0 |
| PI-FGSM | 0.1 | 99.7 | 0.2 | 0.0 | 0.0 | 0.1 | 0.1 | 0.1 | 0.0 | 0.1 | 0.2 | 0.1 | 92.3 |
| $S^2$I-FGSM | 2.0 | 91.4 | 1.9 | 0.5 | 0.8 | 0.0 | 0.3 | 0.1 | 0.0 | 0.0 | 0.0 | 0.6 | 82.9 |
| DiffAttack($1e^{-2}$) | 0.3 | 61.4 | 0.3 | 0.0 | 0.2 | 0.1 | 0.2 | 0.0 | 0.2 | 0.0 | 0.0 | 0.1 | **74.7** |
| DiffAttack($1e^{-1}$) | 6.1 | 99.8 | 5.6 | 3.5 | 6.3 | 4.1 | 3.8 | 2.6 | 3.2 | 0.9 | 1.1 | **3.7** | 147.2 |

Table 16: **Limitation in terms of time and GPU memory consumption.** We report the time for crafting adversarial examples of different attack methods, together with the maximum memory cost. For GAN-based methods (BIA), the adversarial examples are crafted by inferencing the trained generator. While for other methods, the adversarial examples are iteratively optimized with Res-50 as the surrogate model.

| Attack | MI-FGSM | DI-FGSM | TI-FGSM | PI-FGSM | $S^2$I-FGSM | PerC-AL | NCF | BIA | DiffAttack |
|---|---|---|---|---|---|---|---|---|---|
| Mem(MB) | 302 | 336 | 301 | 412 | 305 | 197 | 374 | 242 | 14083 |
| Time(s) | 0.2 | 0.2 | 0.2 | 0.6 | 5.5 | 44.8 | 18.6 | 0.01 | 29.9 |

## L    DISCUSSIONS ON DIFFATTACK'S PERFORMANCE IN TRANSFERABLE TARGETED ATTACK

In this section, we assess the performance of DiffAttack when employed as a targeted attack method. Originally designed for the untargeted attack, we adapt DiffAttack for the targeted task by removing $L_{transfer}$ in Section 3.3 directly. To transform all compared methods in the main paper into targeted attacks, we modify their loss functions by reversing the sign of the classification loss to maximize the logit for the target category. Notably, we exclude PerC-AL as it doesn't support adaptation to targeted attacks, and NCF due to its extremely low success rate in targeted attacks. For target categories, we employ the labels provided in the ImageNet-Compatible Dataset. Adversarial examples are crafted using VGG-19, and the results are presented in Table 15. Differing from the results presented in other tables, here we present the *attack success rate* of the target attacks for clarity. The attack success rate is essentially the complement of the top-1 accuracy, calculated as 100% minus the top-1 accuracy.

From the results, we observe that all models struggle to achieve transferability to black models, a notable and promising avenue for future research. Additionally, when compared to pixel-based attacks, DiffAttack exhibits a lower success rate on the targeted model. We attribute this difference to the tendency of pixel-based attacks to overfit by introducing high-frequency noise. In contrast, unrestricted attacks like DiffAttack often emphasize large-scale patterns with high-level semantics, making it challenging to achieve a high success rate in white-box attacks (this also occurs among those GAN-based attacks). It's worth noting that by increasing the learning rate from $1e^{-2}$ (our default setting) to $1e^{-1}$, DiffAttack can improve its white-box attack success rate and also its transferability. However, this enhancement comes at the cost of reduced fidelity which may be less meaningful.

## M    DISCUSSIONS ABOUT LIMITATION OF TIME AND MEMORY COST

Due to the iterative characteristic and the substantial number of parameters in diffusion models, DiffAttack has a limitation in terms of time and memory consumption compared to other attack methods. In Table 16, we display a comprehensive comparison of computational cost and runtime among DiffAttack, pixel-based attacks, and GAN-based attacks.

The comparison is to process a 224×224 image on a single RTX 3090 GPU. The results reveal that DiffAttack consumes greater memory and generally takes longer to generate adversarial examples.

Table 17: **Demonstration of the effect of the diffusion model itself in enhancing transferability.** We report top-1 accuracy(%). We craft adversarial examples on Inc-v3. "AVG(w/o self)" denotes the average accuracy on all the target models except the ones that have a gray background. The best result is bolded. The first table displays the performance on normally trained models, while the second one on defensive models.

| Ablation | CNNs | | | | | Transformers | | | | MLPs | | AVG(w/o self) | FID |
|---|---|---|---|---|---|---|---|---|---|---|---|---|---|
| | Res-50 | VGG-19 | Mob-v2 | Inc-v3 | ConvNeXt | ViT-B | Swin-B | DeiT-B | DeiT-S | Mix-B | Mix-L | | |
| w/o Diffusion Model | 62.5 | 60.3 | 56.9 | 0 | 88.3 | 85.9 | 87.6 | 88.2 | 84.8 | 68.0 | 62.8 | 74.5 | 69.2 |
| w/o $L_{transfer}$ | 60.6 | 59.2 | 57.4 | 10.9 | 77.9 | 75.1 | 74.4 | 75.2 | 71.9 | 58.6 | 54.7 | **66.5** | **61.6** |

| Ablation | Defensive Models | | | | AVG |
|---|---|---|---|---|---|
| | Adv-Inc-v3 | Inc-v3$_{ens3}$ | Inc-v3$_{ens4}$ | IncRes-v2$_{ens}$ | |
| w/o Diffusion Model | 66.4 | 62.7 | 63.7 | 79.1 | 68.0 |
| w/o $L_{transfer}$ | 45.0 | 43.0 | 42.3 | 57.1 | **46.9** |

This could hinder its deployment in resource-constrained settings, such as autonomous driving and edge models, or for targeting real-time systems.

Notably, this is a common drawback shared by all approaches relying on diffusion models. However, we hope to note that due to the popularity of diffusion models these years, famous communities such as PyTorch and Huggingface keep advancing the efficiency and memory optimization of diffusion models (like Pytorch 2.0 and Diffusers repository). Many recent works (Ulhaq et al., 2022; Dao et al., 2022) have also been dedicated to accelerating diffusion models and addressing memory costs. We firmly believe that these efforts will help bridge the computational gap between DiffAttack and other attack methods in the future, further fostering research on diffusion-based attacks.

## N  FURTHER ABLATION STUDY: ASSESSING THE IMPACT OF THE DIFFUSION MODEL ITSELF ON TRANSFERABILITY AND IMPERCEPTIBILITY

As highlighted in Section 1, the transferability of DiffAttack is not solely attributed to $L_{transfer}$, but also originates from our latent space perturbation and the denoising process intrinsic to the diffusion model itself. In other words, the diffusion model's structure and mechanisms inherently contribute to improving transferability.

To empirically validate this point, we conducted an ablation study by eliminating the diffusion model and directly perturbing the image pixels. This resulted in a pixel-based attack similar to I-FGSM (Kurakin et al., 2017). We aligned the number of iterations with DiffAttack and, to mitigate the generation of unnatural high-frequency noise inherent in pixel-based attacks (as illustrated in Figure 1 in the main paper), we adopted settings from $L_p$-norm-based transferable attacks, limiting the maximum perturbation to 16. To effectively illustrate the influence of the diffusion model itself on transferability, particularly the latent space perturbation and denoising process, we compared this modified degradation model with an adapted DiffAttack (without $L_{transfer}$). We evaluated performance on both normally trained models and four defensive models (Adv-Inc-v3, Inc-v3$_{ens3}$, Inc-v3$_{ens4}$, and IncRes-v2$_{ens}$), yielding the results in Table 17.

The presented results demonstrate that the diffusion model itself can enhance the transferability of adversarial examples, not only on traditionally trained models but also on defensive models. This strongly supports our assertion that the latent space perturbation and defensive denoising process in DiffAttack contribute to improved transferability. Additionally, with the diffusion model, the adversarial examples exhibit lower perceptibility (as indicated by FID scores), further substantiating the motivations outlined in Section 1 and reinforcing the contributions of our work.

## O  MORE QUANTITIVE STUDIES AND VISUALIZATIONS

As a supplement to the experiments in Section 4, we here reveal more experimental results about the parameter settings and also display more visualized comparisons.

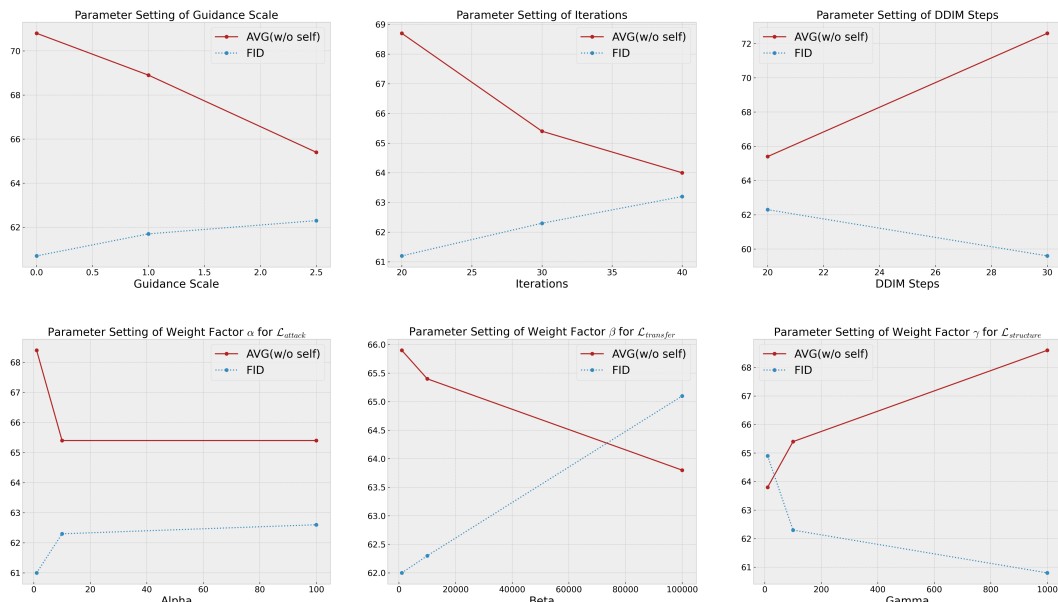

Figure 7: **The effect of different parameter settings.** We conduct a quantitative study on the parameter settings of the guidance scale, iterations, DDIM steps, and weight factors of each loss. "AVG(w/o self)" denotes the average accuracy on all the target models except the one that same as the surrogate one.

**Settings of Guidance Scale.** From Figure 7, it can be observed that with the guidance scale increased, the transferability improves while the imperceptibility deteriorates. We infer this is because larger guidance scales will tend to change the latent more and thus potentially generate more perturbations. Since there is a large gap in the attack success between the guidance scale set to 1.0 and 2.5, but a slight change of the FID value, we set the guidance scale to 2.5 finally.

**Settings of Iterations.** We can notice from Figure 7 that more iterations will sacrifice image quality for the attack success. As more iterations will consume longer optimization time, we here set the number of iterations to 30, which strikes a balance between time-consuming, image quality, and attack robustness.

**Settings of DDIM Steps.** In Figure 7, we keep the DDIM Inversion steps the same (5 inversion steps), to see the effect of different DDIM full sample steps. We do not show here the results for the step number set to 10 because the image quality is rather poor and the structure is completely changed. From the results, we can see that the step number does impact a lot both the transferability and the imperceptibility. Here we set the number of DDIM sample steps to 20, which can produce perceptually invisible adversarial samples with stronger attack robustness.

**Settings of Weight Factor for Loss.** We also conduct quantitative studies on the weight factor settings in Eq. 6 in the main paper. From Figure 7, it can be noticed that our designs of $\mathcal{L}_{transfer}$ and $\mathcal{L}_{structure}$ do make sense for improving the attack's transferability and preserving the content structure. For $\mathcal{L}_{attack}$, we can see from the results that there is a negligible performance improvement when $\alpha$ is increased to a certain extent, thus we set $\alpha$ to 10. For $\mathcal{L}_{transfer}$ and $\mathcal{L}_{structure}$, to balance both the transferability and the imperceptibility, we set them to 10000 and 100 respectively.

**More Visualizations.** We display more visual comparisons in Figure 8 and Figure 9, from which it can be observed that the adversarial examples crafted by our attack are human-imperceptible and hard to be perceived.

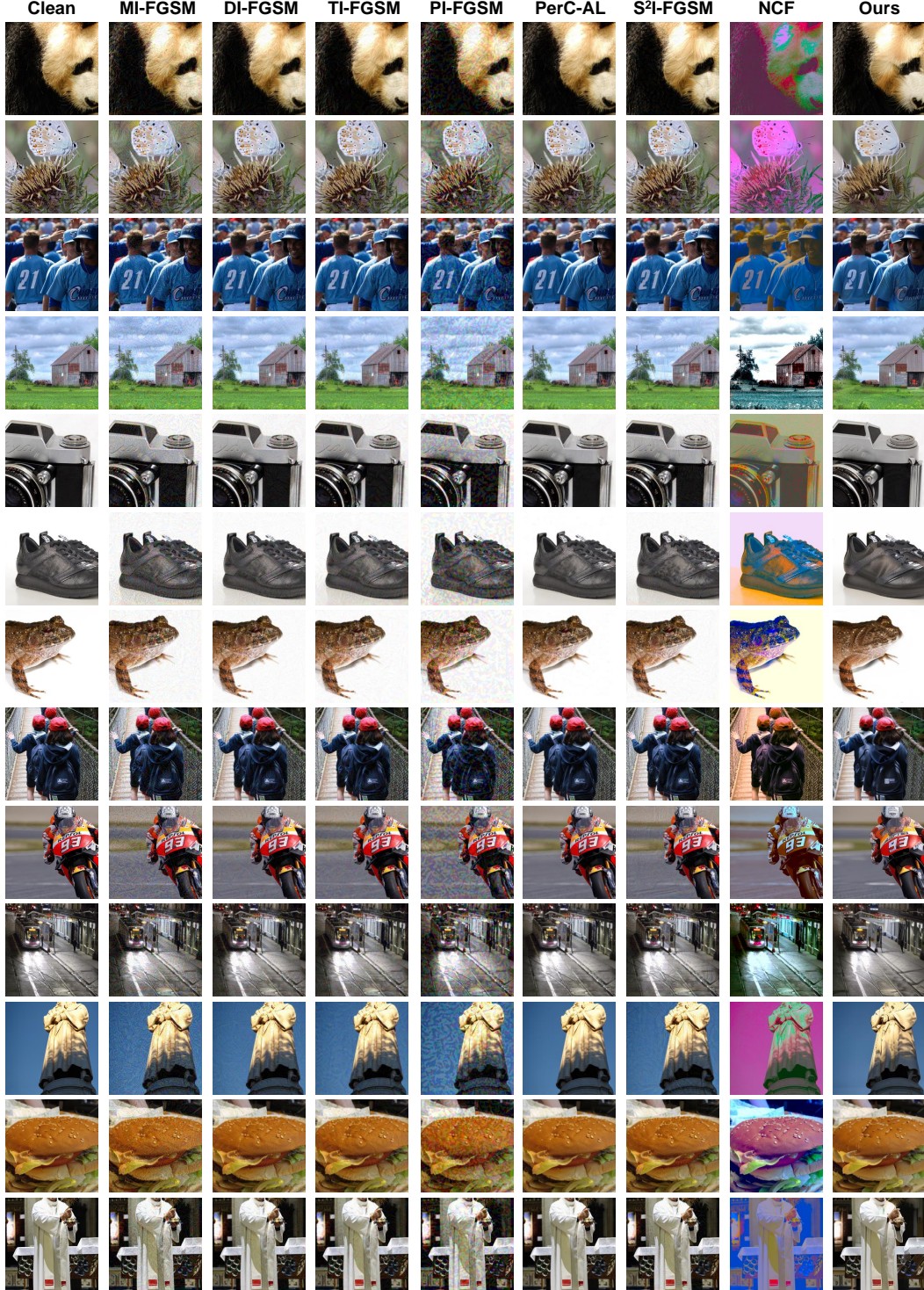

Figure 8: **Supplement visualization of adversarial examples crafted by different attacks.** Please zoom in for a better view.

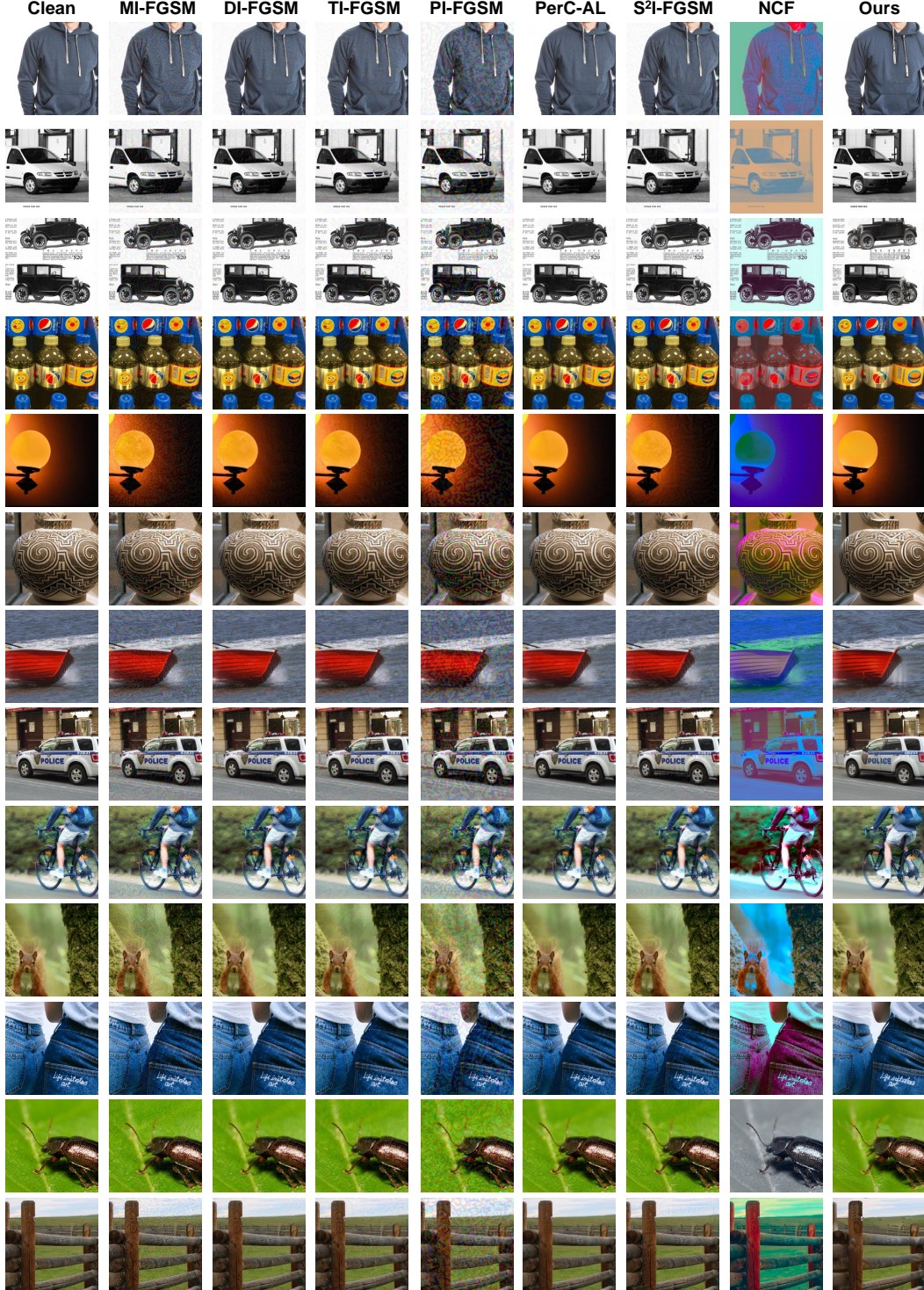

Figure 9: **Supplement visualization of adversarial examples crafted by different attacks.** Please zoom in for a better view.

