# OpenReview forum: "Diffusion Models for Imperceptible and Transferable Adversarial Attack"
_ICLR.cc/2024/Conference — ICLR 2024 Conference Withdrawn Submission_

### Official Review · Reviewer_eRtP · 2023-10-21

**Soundness:** 3 good
**Presentation:** 3 good
**Contribution:** 3 good
**Rating:** 6
**Confidence:** 4

**Summary:**

This paper proposes a novel imperceptible and transferable attack by leveraging both the generative and discriminative power of diffusion models. Different from the existing unrestricted attacks (i.e., manipulation in pixel space), this paper crafts perturbations in the latent space of diffusion models.

**Strengths:**

- The structure of the paper is well organized and most of the descriptions are very clear.
- The performance of the proposed method are powerful, which significantly boosts the transferability and with a good FID.
- The design of the unrestricted attack is intersting, and the resulting adversarial examples looks natural.

**Weaknesses:**

- The comparison of papers may not be so fair. This proposed method additionally relies on stabe diffsion to enhance the transferability, while other methods can only rely on the substitute models.

**Questions:**

- Is the method in this paper time-consuming? Could you provide the time cost, e.g., DiffAttack vs. (NCF, PerC-AL and S$^2$I-FGSM)

---

> ### Author Response · Authors · 2023-11-17
> **Response to Reviewer eRtP**
>
> Thank you sincerely for taking the time to review our work. We extend our heartfelt gratitude for your positive evaluation of our research, which not only encourages us but also instills confidence in our efforts.
>
> ---
>
> >**W1:** The comparison of papers may not be so fair. This proposed method additionally relies on stabe diffsion to enhance the transferability, while other methods can only rely on the substitute models.
>
> Thank you for your insightful concern. We appreciate your consideration of potential fairness issues in comparing our method, given its reliance on stable diffusion to enhance transferability.
>
> To comprehensively address this concern, our manuscript included a thorough analysis and extensive experiments in Appendix K. This analysis was explicitly referenced in the last sentence of Sec. 3.3, where we mentioned, "… DiffAttack exhibits an implicit ensemble characteristic … we give a detailed analysis in Appendix K".
>
> Within Appendix K, we not only demonstrate the fairness of the comparison but also compare our method with other attack approaches that incorporate an additional CLIP classifier. This supplementary analysis aims to further support and validate the effectiveness of our approach.
>
> Furthermore, our superior performance over GAN-based attacks that leverage additional GAN generators, as presented in Appendix J, contributes additional evidence supporting the effectiveness of our approach.
>
> We trust that these points effectively address your concerns. Thank you.
>
> ---
>
> >**Q1:** Is the method in this paper time-consuming? Could you provide the time cost, e.g., DiffAttack vs. (NCF, PerC-AL and SI-FGSM)
>
> Thank you for your insightful advice. We would like to acknowledge that this matter was indeed discussed in the final paragraph of Sec. 5, and a detailed comparison of computational and time costs, along with corresponding analysis, can be found in Appendix M. We hope this information suffices for your inquiry.
>
> We hope that these responses effectively address your questions. Thank you for your continued support.

---

### Official Review · Reviewer_9mVi · 2023-10-22

**Soundness:** 2 fair
**Presentation:** 3 good
**Contribution:** 2 fair
**Rating:** 3
**Confidence:** 3

**Summary:**

The paper proposes a transferable adversarial attack using a diffusion model for stealthiness. They specifically optimized three loss to deceive the classifier while maintaining stealthiness.

**Strengths:**

1) Evaluation results look promising.

2) The writing is clear.

**Weaknesses:**

1) transfer loss: It is confusing that eq.4 optimizes the objective of variance, which is not differentiable. According to the description, it optimizes the objective of evenly distributed cross-attention maps. It disturbs the attention recognition of the diffusion process, but it is unclear how the corruption can be transferred to other pure classifiers without diffusion. Briefly, the loss can help attack diffusion, but how can it help transferability?

2) structure loss: if the purpose is to maintain structure property over diffusion steps, why not directly impose constraints on edges detected by discontinuity detection algorithms or PCA analysis and so on?

3) Evaluation: Lack of ablation study of coefficient in eq. 6 for understanding the interplay of the designed losses.

4) Overall, I feel the loss design (main contribution of the paper) is not fully explained. Basically, it is not convincing why adversarial examples generated by diffusion have better transferability. It is not clear why the designed loss is important, and ablation is missing.

**Questions:**

Please refer to weakness.

---

> ### Author Response · Authors · 2023-11-17
> **Response to Reviewer 9mVi [1/2]**
>
> Thank you sincerely for taking the time to review our paper. We appreciate the questions you've raised, as they provide us with the opportunity to offer further clarification regarding the motivation behind our network design. Below, we present our responses to your queries, aiming to provide a clearer understanding of the rationale behind our loss design.
>
> ---
>
> >**W1:** transfer loss: It is confusing that eq.4 optimizes the objective of variance, which is not differentiable. According to the description, it optimizes the objective of evenly distributed cross-attention maps. It disturbs the attention recognition of the diffusion process, but it is unclear how the corruption can be transferred to other pure classifiers without diffusion. Briefly, the loss can help attack diffusion, but how can it help transferability?
>
> Thank you for your insightful questions. **Our transfer loss enhances DiffAttack by imbuing it with an ensemble attack-like characteristic.** To offer a clearer understanding, we've organized our response into several key sections:
>
> **About the Differentiability of the Variance:**
>
> We need to clarify that the variance operation is indeed differentiable. The calculation of variance, denoted as $\sigma^2$, is expressed as follows:
>
> $$\sigma^2=\frac{1}{n}\sum_{i=1}^{n}\left(x_i-\mu\right)^2$$
>
> As both squaring and summation are differentiable operations, the variance calculation, being a composition of differentiable functions, is itself differentiable.
>
> **About the Benefit to Transferability of This Loss:**
>
> To address concerns about the benefit of our loss design to transferability, two key findings must be acknowledged first:
>
> **Finding 1:** An ensemble-based attack, where adversarial examples are generated on multiple models, enhances transferability compared to a single-model attack. This principle is well-established in prior works such as [1] and [2].
>
> **Finding 2:** The diffusion model, originally designed for image synthesis, implicitly possesses recognition capabilities when pretrained on extensive data like Stable Diffusion. This is highlighted in Para. 2, Sec. 3.3, and validated by [3] and [4].
>
> Building on these findings, our design motivation is to leverage the diffusion model as an implicit surrogate recognition model. Alongside the explicit surrogate recognition model, we craft adversarial examples, making our DiffAttack operate akin to an ensemble attack. Drawing on Finding 1, this ensemble characteristic contributes to improved transferability. We elaborate on this concept in the final sentence of Sec. 3.3, and in Appendix K, we provide a detailed discussion of our method’s ensemble characteristic. The ablation study results in Table 3 affirm the effectiveness of this loss design on transferability, as evidenced by the following excerpt:
>
> |Transfer Loss|Average Accuracy among Black-Box Models|
> |:---:|:---:|
> |w/o|66.5|
> |w/|**65.4**|
>
> **Others:**
>
> Addressing the reviewer's mention of “how the corruption can be transferred to other pure classifiers without diffusion”, we suspect there might be some ambiguity regarding the transferable attack task. In a transferable attack, the attacker crafts adversarial examples on a known recognition model (usually named the surrogate model, distinct from the target model being attacked). These crafted adversarial examples are then expected to deceive other unknown recognition models.
>
> In our method, both the explicit classifier and the implicit diffusion model serve as the surrogate model. We craft adversarial examples based on them and can directly utilize the crafted examples to attack other pure classifiers. There is no requirement for these other classifiers to incorporate diffusion.
>
> We hope these clarifications provide a comprehensive understanding of our design. Thanks.
>
> [1] Liu, Yanpei, et al. "Delving into Transferable Adversarial Examples and Black-box Attacks." International Conference on Learning Representations. 2016.
>
> [2] Dong, Yinpeng, et al. "Boosting adversarial attacks with momentum." Proceedings of the IEEE conference on computer vision and pattern recognition. 2018.
>
> [3] Xu, Jiarui, et al. "Open-vocabulary panoptic segmentation with text-to-image diffusion models." Proceedings of the IEEE/CVF Conference on Computer Vision and Pattern Recognition. 2023.
>
> [4] Clark, Kevin, and Priyank Jaini. "Text-to-Image Diffusion Models are Zero-Shot Classifiers." ICLR 2023 Workshop on Mathematical and Empirical Understanding of Foundation Models. 2023.

---

> > ### Author Response · Authors · 2023-11-17
> > **Response to Reviewer 9mVi [2/2]**
> >
> > >**W2:** structure loss: if the purpose is to maintain structure property over diffusion steps, why not directly impose constraints on edges detected by discontinuity detection algorithms or PCA analysis and so on?
> >
> > Thank you for your question. We appreciate your suggestion to explore alternative methods for structure preservation. While traditional techniques such as edge detection or PCA analysis could be considered, we want to highlight potential challenges associated with their implementation.
> >
> > For instance, employing traditional edge detection methods like Canny may introduce the issue of gradient undifferentiability, impeding end-to-end optimization. Even if we resort to approximated differentiable algorithms, they might compromise optimization performance. On the other hand, incorporating a learnable edge detection model could introduce additional training costs.
> >
> > In contrast, our proposed structure loss presents an innovative and streamlined solution. Leveraging the self-attention map within the diffusion model, our approach ensures internal operations without introducing external constraints. This internalized process enhances the elegance and user-friendliness of our method.
> >
> > We acknowledge the value of your suggestion, and exploring traditional feature-based schemes for structure constraints could be considered in future research. We hope this clarifies the rationale behind our approach. Thank you for your thoughtful comment.
> >
> > ---
> >
> > >**W3:** Evaluation: Lack of ablation study of coefficient in eq. 6 for understanding the interplay of the designed losses.
> >
> > Thank you for your comment. It appears there might have been an oversight during the reviewing process. The ablation study and analysis of the coefficient in Eq. 6 were detailed in Appendix O, and guidance for readers was provided in the last paragraph of Section 4.3, stating, "For more ablation studies on parameter settings, please refer to Appendix O".
> >
> > For your convenience, we have adapted the ablation results into a table below, showcasing the impact of varying $\alpha$, $\beta$, and $\gamma$ on model performance. These coefficients represent loss weights, with the default settings being $\alpha=10$, $\beta=10000$, and $\gamma=100$, as outlined in Sec. 4.1. The results demonstrate the effectiveness of our transfer loss and structure loss in improving attack transferability and preserving content structure.
> >
> > | attack loss $\alpha$ | transfer loss $\beta$ | structure loss $\gamma$ | Average Accuracy among Black-Box Models | FID |
> > |:------:|:-----:|:-----:|:---:|:---:|
> > | 1 | * | * | 68.4 | 61.0  |
> > | 10 | * | * | 65.4 | 62.3  |
> > | 100 | * | * | 65.4 | 62.6  |
> > |  |  |  |  |  |
> > | * | 1000 | * | 65.9 | 62.0  |
> > | * | 10000 | * | 65.4 | 62.3  |
> > | * | 100000 | * | 63.8 | 65.1  |
> > |  |  |  |  |  |
> > | * | * | 10 | 63.8 | 64.9  |
> > | * | * | 100 | 65.4 | 62.3  |
> > | * | * | 1000 | 68.6 | 60.8  |
> >
> > We hope this answer can effectively addresses your concern.
> >
> > ---
> >
> > >**W4:** Overall, I feel the loss design (main contribution of the paper) is not fully explained. Basically, it is not convincing why adversarial examples generated by diffusion have better transferability. It is not clear why the designed loss is important, and ablation is missing.
> >
> > Thank you for your comment. We've addressed most concerns in our responses to your previous three comments. Here, we'd like to delve further into our contribution:
> >
> > While the "loss design" is a significant aspect, our contribution extends beyond it. Our three designed losses enable the creation of imperceptible and transferable adversarial examples in diffusion’s latent space, marking a novel advancement. However, the broader impact of our work lies in inspiring the research community about the untapped potential of diffusion models in adversarial attack strategies, as highlighted at the forefront of our contribution list in the Introduction.
> >
> > Extensive experiments in both the main paper and appendix have validated the superiority of our diffusion-based attack across various scenarios, ranging from multiple datasets to diverse model structures, ensemble attacks, GAN-based comparisons, and resilience to defenses. This comprehensive evaluation positions the diffusion-based attack as a promising research direction. To aid the community in understanding this research potential, we've thoroughly discussed current limitations, explored avenues, and outlined valuable future research directions in Sec. 5.
> >
> > We firmly believe this work will significantly benefit and guide future research in this domain. We kindly request a reevaluation of the contribution and significance of our work. Thank you for your consideration.
> >
> > Please inform us if these clarifications sufficiently address your current questions. We are also more than willing to engage in further discussion about any other unclear aspects. Many thanks for your time and consideration.

---

> ### Author Response · Authors · 2023-11-22
> **Seeking Your Feedback on Our Paper's Rebuttal**
>
> Dear Reviewer 9mVi,
>
> We sincerely appreciate your time and dedication to reviewing our paper. Recognizing the demands of this busy period, we are reaching out to kindly request your feedback on our rebuttal, as the discussion phase nears its conclusion.
>
> If you have any additional comments or suggestions regarding our paper, we would be more than happy to engage in further discussion with you.
>
> Looking forward to your response.
>
> With deepest gratitude,
>
> The Authors

---

### Official Review · Reviewer_WbRq · 2023-11-01

**Soundness:** 3 good
**Presentation:** 3 good
**Contribution:** 2 fair
**Rating:** 5
**Confidence:** 4

**Summary:**

This work studies transferable and imperceptible adversarial attacks using diffusion models. Instead of crafting adversarial perturbations in the pixel space, the authors proposed to use the latent space of diffusion models. Cross-attention and self-attention were also utilized in the attack. Experiments with several baselines showed that the attack can achieve high attack success rates and imperceptibility.

**Strengths:**

- The crating of adversarial perturbations in the latent space of diffusion models is quite intuitive and the paper is easy to follow.
- The experimental results in Table 1 showed that the method can generally achieve good performance wrt a set of baselines.

**Weaknesses:**

- The method seems to have adopted the diffusion model to craft adversarial perturbations in a relatively straightforward way since crafting adversarial attacks in the latent space of generative models has existed before (the diffusion model is a new representative of generative models). Thus it hinders the technical novelty of this work.
- In Table 2, the results show that the method performed worse in attacking NIP-r3 and RS, and it performed worse than PI-FGSM to attack Adv-Inc-v3.
- In Table 4, the authors used FID to measure imperceptibility which is not a visual quality metric, while LPIPS should be more accurate and these results should be presented in the main paper rather than in Appendix.

**Post rebuttal**

I thank the authors for preparing the rebuttal. I have carefully checked the rebuttal and comments from other reviewers, I still feel the technical contributions and the presentation can be further improved. Therefore, I would like to maintain my rating.

**Questions:**

See Weakness.

**Details Of Ethics Concerns:**

NA.

---

> ### Author Response · Authors · 2023-11-17
> **Response to Reviewer WbRq [1/2]**
>
> Thank you for taking the time to review our work and for providing valuable feedback. We are grateful for your acknowledgment of the clarity of our paper and the performance of our method. In response to the highlighted weaknesses, we provide the following clarifications. We trust that these responses will enhance the understanding of the value of our work and address your concerns effectively.
>
> ---
>
> >**W1:** The method seems to have adopted the diffusion model to craft adversarial perturbations in a relatively straightforward way since crafting adversarial attacks in the latent space of generative models has existed before (the diffusion model is a new representative of generative models). Thus it hinders the technical novelty of this work.
>
> Thank you for your insightful feedback. We value this opportunity to address your concerns and **would like to underscore the uniqueness, novelty, and significance of our method to the research community**. Our response is organized into key subparts for clarity:
>
> **Uniqueness of Our Method:**
>
> As depicted in Fig. 2, our approach involves projecting input images into the latent space of diffusion models and perturbing the latent codes to craft adversarial examples. The novel concept of achieving transferable, visually imperceptible adversarial examples by directly manipulating latent space variables of a generative model distinguishes our work. We have pioneered this new paradigm for adversarial attacks, striking a balance between visual imperceptibility and transferability.
>
> The works most closely related to ours are [1,2], which optimize a pre-trained GAN generator’s latent code to craft physical adversarial patches. These crafted patches exhibit significantly different patterns from their background. In contrast, our method crafts imperceptible adversarial examples that do not alter the appearance of the original input image.
>
> [1] Hu, Yu-Chih-Tuan, et al. "Naturalistic physical adversarial patch for object detectors." Proceedings of the IEEE/CVF International Conference on Computer Vision. 2021.
>
> [2] Hu, Zhanhao, et al. "Adversarial texture for fooling person detectors in the physical world." Proceedings of the IEEE/CVF conference on computer vision and pattern recognition. 2022.
>
> **Technical Novelty of Our Method:**
>
> Within our methodology (Sec. 3), our approach consists of three integral components: Basic Framework (Sec. 3.2), "Deceive” Diffusion Model (Sec. 3.3), and Preserve Content Structure (Sec. 3.4). These segments incorporate the design of three distinct loss functions, $L_{attack}$, $L_{transfer}$, and $L_{structure}$.
>
> Regarding the reviewer's comment on the existence of crafting attacks in the latent space of generative models, the focus appears to center solely on the Basic Framework (corresponding to $L_{attack}$). However, our innovation extends beyond this component. Our $L_{transfer}$ and $L_{structure}$ designs, highlighted in the "Deceive” Diffusion Model and Preserve Content Structure sections, enable the creation of transferable and imperceptible adversarial examples within the diffusion model’s latent space. These designs leverage the inherent attention maps of the diffusion model, presenting a distinct approach. The evidence from our ablation experiments and visualizations (Table 3, 4, and Figure 5) underscores the significance of these designs.
>
> We kindly request the reviewer to consider the entirety of our method, encompassing all three designs, while reassessing the novelty of our work, as these innovative components collectively contribute to the advancement in this domain.
>
> **Significance to Community of Our Method:**
>
> Going beyond the specific designs of our method, we emphasize the paramount contribution of our work — the demonstration of diffusion models as a promising foundation for crafting adversarial examples. This pivotal aspect is highlighted as the first contribution in the Introduction's contribution list. Our diffusion-based approach showcases promising outcomes across diverse datasets (Appendix I), various model structures (Sec. 4.2.1 & Appendix H), and demonstrates resilience to defense models (Sec. 4.2.2). Additionally, our method also exhibits superior performance when compared to ensemble attacks (Appendix G & K) and GAN-based methods (Appendix J).
>
> These results collectively underscore the diffusion model as a robust platform for crafting adversarial examples. Beyond empirical validation, our work offers a comprehensive exploration of potential future directions for diffusion-based methods in the discussion section (Sec. 5). We believe that this exploration not only enhances the understanding of diffusion-based models' potential within the community but also provides valuable guidance for future research endeavors.
>
> In light of these clarifications, we believe our work stands as a very valuable contribution to the field and hope you find it equally compelling after these explanations. Many thanks.

---

> > ### Author Response · Authors · 2023-11-17
> > **Response to Reviewer WbRq [2/2]**
> >
> > >**W2:** In Table 2, the results show that the method performed worse in attacking NIP-r3 and RS, and it performed worse than PI-FGSM to attack Adv-Inc-v3.
> >
> > Thank you for your meticulous observation. **The observed suboptimal performance of our method against specific defensive models can be traced back to the choice of the surrogate model—specifically, the relatively weaker-performing Inc-v3.** In response to your concerns, we delve into two key aspects: an analysis of the suboptimal performance and an elucidation of our rationale behind showcasing results based on Inc-v3:
> >
> > **Analysis of the Suboptimal Performance:**
> >
> > It's important to acknowledge that no single method universally excels in all scenarios. Table 1 and 9 showcase our method's substantial advantages across various model structures, including ResNet-50, VGG-19, MobileNet-v2, ViT-B, Mixer-B, etc. Notably, on structures like VGG-19 and MobileNet-v2, our method outperforms the second-best attack approach by nearly 10 points.
> >
> > However, we recognize the challenges our method faces under specific model structures, particularly Inc-v3, where its transferability ranks 3rd among other attack methods. As we craft adversarial examples based on Inc-v3 for assessing defensive resilience in Table 2, this suboptimal transferability directly influences its resilience against certain defense methods (NIP-r3, RS, Adv-Inc-v3) employing multiple or different model structures.
> >
> > Despite these challenges, it's crucial to highlight that, even when based on the least performing model structure (Inc-v3), our method still achieves the best resilience among numerous other defensive methods in Table 2. This underscores the overall efficacy of our approach, emphasizing its robustness in diverse scenarios.
> >
> > To further support this clarification, we present the performance of our method when crafting adversarial examples on the "better" VGG structure. The favorable results achieved below underscore the robustness of our approach:
> >
> > (“VGG-19$_{normal}$'' is the accuracy on normally trained VGG-19. For NRP and DiffPure, we display the accuracy differences after purification. The best result is bolded, and the second-best result is underlined.)
> > |Attack\Defense|HGD|R\&P|NIP-r3|RS|Adv-Inc-v3|Inc-v3$_{ens3}$|Inc-v3$_{ens4}$|IncRes-v2$_{ens}$|VGG-19$_{normal}$|NRP|DiffPure|
> > |:-:|:-:|:-:|:-:|:-:|:-:|:-:|:-:|:-:|:-:|:-:|:-:|
> > |MI-FGSM|78.0|61.9|83.2|65.1|62.2|61.2|64.5|77.7|0.0|+4.2|+49.7|
> > |DI-FGSM|82.3|76.9|88.9|71.1|70.5|67.2|69.5|83.7|0.0|+18.7|+61.1|
> > |TI-FGSM|81.6|79.8|87.5|68.7|69.9|65.9|68.1|80.8|0.0|+21.6|+57.3|
> > |PI-FGSM|78.8|$\underline{58.3}$|73.6|35.3|$\underline{48.9}$|52.1|53.7|67.8|0.0|+6.0|+14.3|
> > |S$^2$I-FGSM|69.6|64.0|78.8|65.1|61.2|56.7|60.4|72.0|0.0|+3.4|+54.3|
> > |PerC-AL|95.8|96.5|94.6|74.0|81.5|76.4|76.9|88.4|4.6|+53.5|+65.5|
> > |NCF|$\underline{67.2}$|72.3|$\underline{64.6}$|$\underline{28.8}$|49.0|$\underline{46.4}$|$\underline{47.7}$|$\underline{60.5}$|6.8|+13.3|+13.2|
> > |DiffAttack(Ours)|**46.3**|**45.7**|**51.3**|**28.2**|**36.9**|**38.6**|**37.6**|**48.7**|2.7|**+2.2**|**+12.8**|
> >
> > **The reason why we display results on Inc-v3:**
> >
> > Regarding our choice to display results for Inc-v3 instead of other models with better outcomes in Table 1, our intention is to showcase the lower boundary of our method's performance. While presenting state-of-the-art results is inherently satisfying, our decision reflects a strategic choice. By demonstrating outcomes on Inc-v3, a less performant model, we aim to elucidate the method's performance boundaries, providing valuable insights for future research and enhancing the reader's understanding of our method’s capability. We believe this choice contributes significantly to a broader comprehension of our method for potential interested readers.
> >
> > We hope this response effectively addresses your concerns. Thank you.
> >
> > ---
> >
> > >**W3:** In Table 4, the authors used FID to measure imperceptibility which is not a visual quality metric, while LPIPS should be more accurate and these results should be presented in the main paper rather than in Appendix.
> >
> > Thank you for your guidance. Our initial use of FID aimed to assess the distribution distance between crafted adversarial examples and clean images, determining the realism of our crafted adversarial examples. If these examples fall within a closely aligned distribution, they can be considered imperceptible to humans.
> >
> > We appreciate your constructive suggestion to enhance Table 4 by including the LPIPS metric, providing readers with a clearer perception of our performance. Here are the updated results for Table 4 with the additional LPIPS metric:
> >
> > |Steps Inversed|Self-Attn Control|Initial Recon|FID|LPIPS|
> > |:-:|:-:|:-:|:-:|:-:|
> > |10|-|-|97.9|0.3719|
> > |5|-|-|66.7|0.1424|
> > |5|+|-|63.5|0.1313|
> > |5|+|+|62.3|0.1256|
> >
> > These results will be incorporated into our next revision, and we will also transfer the LPIPS results from the appendix to the main paper. We hope these updates address your concern. Many thanks.

---

> ### Author Response · Authors · 2023-11-22
> **Sincere Request for Reconsideration [Final]**
>
> Dear Reviewer,
>
> Thank you for your response to our rebuttal. **We appreciate the time and consideration you've dedicated to evaluating our submission**.
>
> We want to express our genuine frustration with the current evaluation outcome. It's disheartening to feel that our efforts in the previous rebuttal might not have been fully acknowledged, although we understand this can be a common occurrence in conference reviews. Nonetheless, **rather than resigning to disappointment, we'd like to make a final effort to appeal for a Rating Increase**.
>
> Allow us to present some additional insights for your consideration and that of potential readers, articulated in bullet points for clarity:
>
> - Our work stands as the **pioneering venture that introduces diffusion models into the realm of adversarial attacks**, garnering relatively **good attention within the field**.
> - We have **an array of experiments within the manuscript to bolster its comprehensiveness**. This ICLR submission showcases **numerous experiments and comparisons**, with **our method achieving the best results in various situations**.
>
> We're perplexed by the notion that our work lacks novelty. Being **the first** to delve into **diffusion-based attacks**, proposing **effective designs** of utilizing diffusion models, conducting **extensive experiments and comparisons** yielding **promising results**, and **delving deeply into the future research direction** of diffusion-based attacks—**Don't these elements establish the novelty and value of our work?**
>
> On the other hand, **the word "feel"** in the reviewer's reply to our rebuttal is, to be honest, **subjective and ambiguous**. **We genuinely hope that the reviewer can carefully reassess our contribution**.
>
> **While we understand that perspectives on novelty can vary, we are eager for another opportunity to advocate for our work.** We earnestly wish for this work to be shared at the conference, allowing more individuals to recognize the potential of diffusion-based attacks.
>
> Thank you for your time and consideration.
>
> Best regards and wishes for a good day,
>
> Authors of Submission 2148

---

### Author Response · Authors · 2023-11-21
**Look forward to your responses!**

Dear reviewers,

We wish to convey our sincere gratitude once again for the invaluable time and effort you have dedicated to reviewing our submission.

As we approach the end of the rebuttal period, we kindly inquire whether our responses have effectively addressed your concerns or questions. We are committed to addressing any remaining concerns with utmost eagerness.

We sincerely look forward to your responses.

Respectfully,

Authors